# *Some Optimizers are More Equal:* Understanding the Role of Optimizers in Group Fairness

**Mojtaba Kolahdouzi[1], Hatice Gunes[2], Ali Etemad[1]**
[1] Department of Electrical and Computer Engineering, Queen's University, Canada
[2] Department of Computer Science and Technology, University of Cambridge, United Kingdom
`m.kolahdouzi@queensu.ca, hatice.gunes@cl.cam.ac.uk, ali.etemad@queensu.ca`

## Abstract

We study whether and how the choice of optimization algorithm can impact group fairness in deep neural networks. Through stochastic differential equation analysis of optimization dynamics in an analytically tractable setup, we demonstrate that the choice of optimization algorithm indeed influences fairness outcomes, particularly under severe imbalance. Furthermore, we show that when comparing two categories of optimizers, adaptive methods and stochastic methods, RMSProp (from the adaptive category) has a higher likelihood of converging to fairer minima than SGD (from the stochastic category). Building on this insight, we derive two new theoretical guarantees showing that, under appropriate conditions, RMSProp exhibits fairer parameter updates and improved fairness in a single optimization step compared to SGD. We then validate these findings through extensive experiments on three publicly available datasets, namely CelebA, FairFace, and MS-COCO, across different tasks as facial expression recognition, gender classification, and multi-label classification, using various backbones. Considering multiple fairness definitions including equalized odds, equal opportunity, and demographic parity, adaptive optimizers like RMSProp and Adam consistently outperform SGD in terms of group fairness, while maintaining comparable predictive accuracy. Our results highlight the role of adaptive updates as a crucial yet overlooked mechanism for promoting fair outcomes. We release the source code at: https://github.com/Mkolahdoozi/Some-Optimizers-Are-More-Equal.

## 1 Introduction

Machine learning is increasingly applied to a wide variety of domains, many of which have significant social implications [5, 43]. These include, but are not limited to, decision-making systems, risk assessment models, recommendation engines, and personalized health interventions [12, 46]. These systems influence people's lives and can lead to unintended biases or disparities [38], for instance by treating different groups of people differently. To address this, researchers have developed various techniques to ensure *group fairness* in machine learning algorithms [20, 13].

Prior work has demonstrated that a variety of factors such as training data imbalance [45] and model hyper-parameters [10] can impact the performance of deep neural networks in terms of group fairness and bias. In this paper, we pose the following question for the first time:

> *Does the choice of optimization algorithm impact group fairness in deep neural networks? and how?*

The limited theoretical and empirical analysis in this area leaves a major gap in our understanding of how optimizers impact the inherent performance of models in terms of bias. Consequently, we believe that this question can have significant implications in developing models that perform consistently across various groups, as optimization is a fundamental component of any deep learning model.

39th Conference on Neural Information Processing Systems (NeurIPS 2025).

In this paper, we bridge this gap by first analyzing the impact of optimization in group fairness in an analytically tractable setup. This initial study confirms our hypothesis: that indeed the choice of optimizer impacts group fairness in downstream applications, particularly in the presence of severe bias in the training set. Next, we delve deep into the problem by proposing and proving two new theorems that demonstrate that among the two primary families of optimization algorithms, adaptive gradient methods such as Adam and RMSProp, and stochastic gradient methods such as SGD, the former is more likely to exhibit better group fairness. Next, we validate our theoretical findings by conducting extensive experiments on three publicly available datasets-MS-COCO [27], CelebA [31], and FairFace [16] across three different tasks: multi-label classification, facial expression recognition, and gender classification. We carefully evaluate fairness using three widely adopted metrics: Equalized Odds, Equal Opportunity, and Demographic Parity. The results are strongly aligned with our theoretical analysis, consistently demonstrating that adaptive gradient methods achieve better fairness scores compared to stochastic gradient methods.

Our contributions in this paper are summarized as follows: (**1**) First, we provide a detailed mathematical analysis to study the impact of optimization algorithms on fairness in an analytically tractable setting. Here, we demonstrate that adaptive gradient algorithms like RMSProp have a higher probability of converging to fairer minima. (**2**) Next, we establish and prove two theorems showing that under appropriate conditions: (*a*) RMSProp provides fairer updates than SGD, and (*b*) the worst-case bias introduced by RMSProp in a single optimization step has an upper bound equal to that of SGD. These theoretical insights highlight key properties of optimization algorithms that contribute to their advantages in terms of fairness. (**3**) Through extensive experiments on multiple publicly available datasets and tasks, we empirically validate our theoretical findings. We systematically evaluate the fairness of models optimized using adaptive gradient methods and compare the outcomes to those obtained from stochastic gradient methods. Our results consistently show that adaptive gradient methods lead to fairer minima. (**4**) To enable rapid reproducibility and contribute to the area, we make our code publicly available at: https://github.com/Mkolahdoozi/Some-Optimizers-Are-More-Equal.

## 2 Related work

**Group fairness.** Bias in machine learning models leads to discriminatory outcomes, and as these models are increasingly being deployed in real-life applications, such biases can impose risks to society [38]. This risk is particularly important in high-stakes applications, such as hiring [17], healthcare [22], etc. When models yield biased outcomes, they favor certain demographic groups, which reinforces social inequalities. As an example, facial expression recognition models show higher error rates for darker skin tones [55]. Such biases undermine trust in machine learning models and pose ethical and legal challenges. To analyze fairness in machine learning, researchers in this area often distinguish the notion of "group fairness" and "individual fairness" [38]. Group fairness focuses on the notion that machine learning models treat different demographic groups, such as those defined by race, the same. However, individual fairness dictates that similar individuals should be treated similarly. In this paper, we focus on *group fairness*, which is more widely recognized in practical applications [7].

One of the main sources of bias in machine learning models is data imbalance [9]. When certain demographic subgroups are underrepresented, models prioritize patterns from the majority group, leading to biased decisions. To tackle this, a variety of methods have been proposed that can be categorized into 3 main groups [6]: pre-processing, in-processing, and post-processing. Pre-processing methods often apply some type of transformation to data to make the demographic information less recognizable to the model. As an example, [48] augments the training set with synthetically generated samples, thereby ensuring an equal distribution for demographic subgroups. In-processing methods directly modify the learning algorithm of the model. As an example, [20] proposes a new loss function based on maximum mean discrepancy statistical distance to enhance fairness. Post-processing methods intervene after training is completed. For instance, [44] introduces a post-processing approach using graph Laplacian regularization to enforce individual fairness constraints while preserving accuracy. Nevertheless, in practice, these fairness-enhancing methods are rarely used in real-world pipelines [39]. One major limitation is their computational overhead or interference with the training phase. As an example, pre-processing methods require extensive data augmentation or transformations. Considering these challenges, an alternative approach to enhance fairness is to use components that are inherently present in every training pipeline. One of

the fundamental building blocks of modern deep learning is optimization [52]. Understanding the role of different optimization algorithms in fairness is crucial for practitioners seeking to develop fairer models without additional fairness-specific modifications. The work in [29] provides a key theoretical analysis, showing that standard and unconstrained machine learning implicitly favors one fairness criterion over others. The authors prove that a model's deviation from calibration is bounded by its excess risk. This is related to our work as it also investigates the fairness properties that emerge from a standard training process. Furthermore, [56] focuses on in-processing fairness methods that use surrogate functions to ensure fairness. Their theory also highlights that balanced datasets are beneficial for achieving tighter fairness and stability guarantees. This is relevant to our findings, as we also identify data imbalance as a critical factor through stochastic differential equation analysis.

**Optimization algorithms.** Optimization algorithms in deep learning generally fall into two main categories: SGD and adaptive gradient methods (e.g., Adam [18] and RMSProp [34]). SGD updates model parameters using a constant learning rate [15], while adaptive methods dynamically adjust per-parameter learning rates based on past gradients, often leading to faster convergence [35]. The impact of these two types of optimization algorithms on group fairness remains not well understood. Nonetheless, several studies have explored their effects in other areas. For instance, [34] empirically demonstrates that models trained with SGD are more robust to input perturbations than those trained with adaptive methods. Fairness and robustness are often competing objectives in deep learning [36]. This hints that adaptive optimizers may yield better fairness compared to SGD. However, this relationship remains largely unexplored.

# 3 Method

## 3.1 Preliminaries

**Optimization algorithms.** Stochastic gradient methods like SGD and adaptive gradient methods such as RMSProp and Adam are widely used for optimizing deep neural networks [58]. Given a loss function $\mathcal{L}(w)$, SGD updates the model parameters $w$ at each iteration $k$ as:

$$w_{k+1} = w_k - \eta \nabla \mathcal{L}(w_k), \tag{1}$$

where $\eta$ denotes the learning rate. Adaptive gradient methods improve the speed of convergence by applying coordinate-wise scaling to the gradient [41]. Specifically, RMSProp normalizes the gradient with an exponentially decaying average of its squared values:

$$v_k = \gamma v_{k-1} + (1 - \gamma) \nabla \mathcal{L}(w_k)^2, \quad w_{k+1} = w_k - \frac{\eta}{\sqrt{v_k + \epsilon}} \nabla \mathcal{L}(w_k), \tag{2}$$

where $v_k$ is the moving average of squared gradients, $\gamma$ is the decay factor, and $\epsilon$ is a small constant added for numerical stability. On the other hand, Adam works by considering both the first and the second moments of gradients:

$$m_k = \beta_1 m_{k-1} + (1 - \beta_1) \nabla \mathcal{L}(w_k), \tag{3}$$

$$v_k = \beta_2 v_{k-1} + (1 - \beta_2) \nabla \mathcal{L}(w_k)^2, \tag{4}$$

$$\hat{m}_k = \frac{m_k}{1 - \beta_1^k}, \quad \hat{v}_k = \frac{v_k}{1 - \beta_2^k}, \tag{5}$$

$$w_{k+1} = w_k - \frac{\eta}{\sqrt{\hat{v}_k} + \epsilon} \hat{m}_k, \tag{6}$$

where $\beta_1$ and $\beta_2$ are the first and second moment decay factors, respectively.

**Stochastic Differential Equations (SDE).** Given the interval $[0, T]$ where $T > 0$, we have [51]:

$$dX_t = a(X(t), t)dt + b(X(t), t)dB(t), \tag{7}$$

where $X_t \in R^d$ is a stochastic process (the solution of the SDE), $a : R^d \times [0, T] \to R^d$ is the drift term, $b : R^d \times [0, T] \to R^{d \times s}$ is the diffusion term, and $B(t) \in R^s$ is $s$-dimensional Brownian motion. Intuitively, SDEs extend ordinary differential equations (ODEs) by incorporating stochastic noise. Analyzing functions involving stochastic processes requires tools beyond classical calculus. A fundamental theorem in stochastic analysis is Itô's lemma, extending the chain rule to the functions

of stochastic processes [1]. Please see the Appendix A for Itô's lemma. Unlike ODEs, the solution of an SDE is a stochastic process $X_t$. Formally, the solution can be represented as:

$$X_t = X_0 + \int_0^t a(X_\tau, \tau)d\tau + \int_0^t b(X_\tau, \tau)dB_\tau. \tag{8}$$

The second integral, also known as the Itô's integral, captures the stochastic contribution driven by Brownian motion. It is defined as:

$$\int_0^t b(X_\tau, \tau)dB_\tau = \lim_{n \to \infty} \sum_{\tau_k, \tau_{k-1} \in \pi(n)} b(X_{\tau_{k-1}}, \tau_{k-1})(B_{\tau_k} - B_{\tau_{k-1}}), \tag{9}$$

where $\pi(n)$ denotes a sequence of $n$ partitions in $[0, t]$, and $\lim$ shows convergence in probability. In many cases, explicitly computing $X_t$ is intractable. Instead, it is often more practical to analyze the probability distribution of $X_t$. The Fokker-Planck equation, also known as the forward Kolmogorov equation, describes the time evolution of the probability density function $p(x, t)$ of $X_t$ [21] as:

$$\frac{\partial p(x, t)}{\partial t} = -\nabla \cdot (a(x, t)p(x, t)) + \frac{1}{2}\nabla \cdot \left(\nabla \cdot (b(x, t)b(x, t)^t p(x, t))\right), \tag{10}$$

where $\nabla\cdot$ represents the divergence operator. For completeness, we provide the derivation of the Fokker-Planck equation from Eq. 7 in Appendix B.

**SDE approximations for optimization algorithms.** SDEs provide a powerful framework for analyzing the dynamics of discrete optimization algorithms such as SGD and RMSProp in a continuous setup. While direct analysis of discrete updates is often challenging [26], SDE approximations enable the study of optimization trajectories. In the context of optimization, the sources of stochasticity include mini-batch gradient noise which are variations due to data heterogeneity (e.g., differences in mini-batches drawn from different subpopulations), among others. To formalize this, we adopt the Noisy Gradient Oracle with Scale Parameter (NGOS) [37], which models stochastic gradients as $g(w) = \nabla\mathcal{L}(w) + \Theta z$, where $\Theta$ is a noise scale parameter, and $z$ is a random variable that follows a distribution with covariance matrix $\Sigma(w)$ and mean of zero. NGOS formally models the gradients used in mini-batch training. It describes the gradient from a small data batch as the "true" gradient (which would come from the full dataset) plus random noise introduced by the sampling process. A well-behaved NGOS is defined as $\nabla\mathcal{L}(w)$ being Lipschitz and $\sqrt{\Sigma(w)}$ being bounded. See Appendix C for a detailed description of a well-behaved NGOS. Note that these assumptions are common and standard in prior works that study optimization using SDEs [25].

Under well-behaved NGOS, SGD can be approximated by the following SDE [25]:

$$dW_t = -\nabla\mathcal{L}(W_t)dt + (\eta\Sigma(W_t))^{1/2}dB(t), \tag{11}$$

where $W_t$ represents an approximation of $w_k$ at discrete time steps $k\,\eta$. More recently, SDE approximation for RMSProp has been proposed as a coupled system of equations [37]:

$$dW_t = -P_t^{-1}\left(\nabla\mathcal{L}(W_t)dt + \Theta\,\eta\,\Sigma^{1/2}(W_t)dB(t)\right), \quad du_t = \frac{1-\gamma}{\eta^2}(diag(\Sigma(W_t)) - u_t)dt, \tag{12}$$

where $P_t = \Theta\,\eta\,diag(u_t)^{\frac{1}{2}} + \epsilon\,\eta\,I$ and $u_t$ is defined as $\frac{v_k}{\Theta^2}$. Note that these SDE approximations converge to $w_k$ in the weak sense, meaning that while individual trajectories of the discrete and continuous processes may differ, their probability distributions remain close. Please see Appendix D for a formal formulation of the convergence.

## 3.2 SDE analysis of optimizers

To illustrate the different behaviors of SGD and RMSProp in terms of group fairness, let's consider a warm-up example. In this example, assume a training set consisting of two equally-sized subgroups, denoted as subgroups 0 and 1. That is, each group constitutes 50% of the total population. Please note this example can be generalized over more than 2 subgroups. For simplicity, assume each subgroup has its own loss function, denoted as:

$$\mathcal{L}_0(w) = \frac{1}{2}(w - 1)^2, \quad \mathcal{L}_1(w) = \frac{1}{2}(w + 1)^2. \tag{13}$$

The objective is to minimize a population-level loss function defined as the weighted sum of the subgroup losses as:

$$\mathcal{L}_{\text{pop}}(w) = 0.5\mathcal{L}_0(w) + 0.5\mathcal{L}_1(w), \tag{14}$$

where the weights correspond to the proportion of each subgroup in the population. In this setup, the optimal solution for subgroup 0 is $w_0^* = 1$, while the optimal solution for subgroup 1 is $w_1^* = -1$. However, The population's optimal solution is obtained by minimizing $\mathcal{L}_{\text{pop}}(w)$, leading to $w_{\text{pop}}^* = 0$. The following lemma shows that under the demographic parity definition of group fairness [38], $w_{\text{pop}}^* = 0$ is the fairest minima.

**Lemma 1.** *Let $\mathcal{L}_0(w) = \frac{1}{2}(w-1)^2$ and $\mathcal{L}_1(w) = \frac{1}{2}(w+1)^2$ be the loss functions for subgroups 0 and 1, respectively. Define the population loss as $\mathcal{L}_{pop}(w) = 0.5\mathcal{L}_0(w) + 0.5\mathcal{L}_1(w)$. Under the demographic parity definition of fairness, the fairest minimizer of $\mathcal{L}_{pop}(w)$ is $w_{pop}^* = 0$.*

*Proof.* See Appendix E. □

In practice, we do not have access to the full population above, rather we have access to the set $\Omega$ of $N$ individual samples, sampled uniformly from the population. Additionally, set $\Omega$ may not contain an equal number of samples from subgroups 0 and 1, thus introducing bias. We assume that the probability of subgroup 0 in $\Omega$ is $p_0$ while the probability of subgroup 1 is $p_1 = 1 - p_0$. For a given set $\Omega$, the empirical minimization problem used in practice is defined as: $\mathcal{L}_{emp}(w) = \frac{1}{N}\sum_{r\in\Omega}\mathcal{L}_{q_r}(w)$, where $q_r \in \{0,1\}$ corresponds to samples from subgroups 0 and 1. Note that as $N \to \infty$, $\mathcal{L}_{emp} \to \mathcal{L}_{pop}$. In min-batch training with batch size of 1, at each iteration, the optimizer processes a single individual from subgroup 0 with a probability of $p_0$ and from subgroup 1 with a probability $p_1$. In other words, with both SGD and RMSProp, the model receives $\nabla\mathcal{L}_0(w)$ with a probability of $p_0$ and $\nabla\mathcal{L}_1(w)$ with a probability of $p_1$, updating $w$ according to Eqs. 1 and 2, respectively. If $p_0 > p_1$, the optimization trajectory is biased towards the group-specific minimum at $w = 1$, and $w = -1$ otherwise. This demonstrates that disproportionate sampling of subgroups leads to unfair minima, which is particularly relevant in practical settings. The following theorem establishes that when the sampling bias $|p_0 - p_1|$ exceeds a certain threshold, RMSProp has a higher probability of converging to $w_{pop}^*$ compared to SGD.

**Theorem 1.** *Let $p_0, p_1 \in (0,1)$ with $p_0 + p_1 = 1$ be the subgroup sampling probabilities for the loss functions $\mathcal{L}_0(w) = \frac{1}{2}(w-1)^2$ and $\mathcal{L}_1(w) = \frac{1}{2}(w+1)^2$. Suppose we optimize the empirical objective $\mathcal{L}_{emp}(w) = \frac{1}{N}\sum_{r\in\Omega}\mathcal{L}_{q_r}(w)$, where each sample $q_r \in \{0,1\}$ is drawn i.i.d. with probability $p_0$ for subgroup 0 or $p_1$ for subgroup 1. Consider mini-batch gradient updates of size 1 (i.e., one sample per iteration) using SGD and RMSProp optimization algorithms. Then there exists a constant $\Delta(p_1 p_2, \eta) > 0$ such that, whenever $|p_0 - p_1| > \Delta(p_1 p_2, \eta)$, we have: $\frac{p_{rms}(w_{pop}^*)}{p_{sgd}(w_{pop}^*)} > 1$, where $p_{rms}(w_{pop}^*)$ and $p_{sgd}(w_{pop}^*)$ are the probabilities of RMSProp and SGD converging to fair minima $w_{pop}^* = 0$, respectively.*

*Proof.* As mentioned earlier, the gradient of $\mathcal{L}_{emp}$ is $\nabla\mathcal{L}_0(w)$ with a probability $p_0$ and $\nabla\mathcal{L}_1(w)$ with a probability of $p_1$. Thus, the covariance is $\Sigma(W_t) = 4p_0 p_1$. This means $\Sigma(W_t)$ is independent of $W_t$. When convergence occurs for RMSProp, $u_t$ defined in Eq. 12 converges to $\Sigma(W_t)$. Thus, $u_t = 4p_0 p_1$. With this in mind, let's derive the SDE for RMSProp:

$$dW_t = \frac{-1}{\eta(2\Theta\sqrt{p_0 p_1} + \epsilon)}(\nabla\mathcal{L}_{pop}(W_t)dt + 2\Theta\eta\sqrt{p_0 p_1}dB_t). \tag{15}$$

Here, $\epsilon$ is added for numerical stability, but it can be ignored as it is commonly done so in prior work [37]. Next, let's derive the SDE for SGD:

$$dW_t = -\nabla\mathcal{L}_{pop}(W_t)dt + 2\sqrt{\eta}\sqrt{p_0 p_1}dB(t). \tag{16}$$

For analyzing the two SDEs above, we use the Fokker-Planck equations, presented in Eq. 10. When convergence occurs, $\frac{\partial p(w,t)}{\partial t} = 0$. Thus, Fokker-Planck for RMSProp can be written as:

$$\frac{\partial}{\partial w}\left(p(w) \times \frac{\nabla\mathcal{L}_{pop}(w)}{2\eta\Theta\sqrt{p_0 p_1}}\right) + \frac{1}{2}\frac{\partial^2 p(w)}{\partial w^2} = 0. \tag{17}$$

Substituting $\nabla \mathcal{L}_{pop} = \frac{p_0}{2}(w-1) + \frac{p_1}{2}(w+1)$ in the equation above, we obtain:

$$\frac{\partial}{\partial w}\left(p(w) \times \frac{p_0(w-1) + p_1(w+1)}{4\eta\Theta\sqrt{p_0 p_1}}\right) + \frac{1}{2}\frac{\partial^2 p(w)}{\partial w^2} = 0. \tag{18}$$

Similarly, the Fokker-Planck equation for SGD can be written as:

$$\frac{\partial}{\partial w}\left(p(w) \times \frac{p_0(w-1) + p_1(w+1)}{2}\right) + 2\eta p_0 p_1 \frac{\partial^2 p(w)}{\partial w^2} = 0. \tag{19}$$

Eq. 18 is an analytically solvable ordinary differential equation, which can be written as:

$$p_{rms}(w) = \sqrt{\frac{\kappa}{\pi}}exp\left(-\kappa(w - (p_0 - p_1))^2\right), \tag{20}$$

where $\kappa = \frac{1}{4\eta\Theta\sqrt{p_0 p_1}}$. As a sanity check, we can see that the expectation of the distribution in Eq. 20 is equal to $p_0 - p_1$. This means in the case of $p_0 = p_1$, i.e. no sampling bias, the weight converges to $w^* = 0$, the unbiased minimum. Similarly, we can write the analytical solution of Eq. 19 as:

$$p_{sgd}(w) = \sqrt{\frac{\vartheta}{\pi}}exp\left(-\vartheta(w - (p_0 - p_1))^2\right), \tag{21}$$

where $\vartheta = \frac{1}{8\eta p_0 p_1}$. As we use a batch size of 1, we can set $\Theta = 1$ [37]. Then, $1 >= 2\sqrt{p_0 p_1}$ inequality holds. With this in mind, by substituting $w^*_{pop} = 0$ in Eqs. 20 and 21, we can conclude $\frac{p_{rms}(w^*_{pop})}{p_{sgd}(w^*_{pop})} > 1$ holds if and only if $|p_0 - p_1| > \Delta(p_1 p_2, \eta)$, where $\Delta(p_1 p_2, \eta) = \sqrt{\frac{\frac{1}{2}\ln\frac{\vartheta}{\kappa}}{\vartheta - \kappa}}$. $\qquad\square$

Now, let's perform a quick simulation to verify our findings. We use the loss functions defined in Eq. 13 and set $p_0 = 0.1$ and $p_1 = 0.9$ to induce a severe sampling bias. We fix the learning rate to $\eta = 0.1$ and apply SGD and RMSProp for 100 epochs. Each experiment is repeated 1000 times, and at every epoch we record the fraction of runs converging to the neighborhood of the fair optimum $w^*_{pop}$, defined by $|w - w^*_{pop}| < 0.2$. The result is shown in Fig. 1 (left). As it is evident from this figure, approximately 10% of the runs under RMSProp lie in the neighborhood of $w^*_{pop}$, whereas SGD

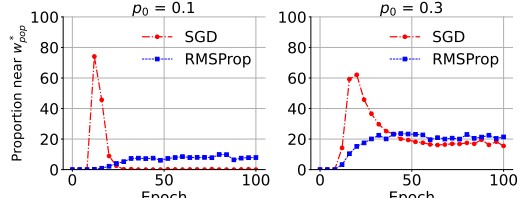

Figure 1: Percentage of 1000 runs converging within the fair neighborhood for SGD and RMSProp, under severe bias ($p_0 = 0.1$) and mild bias ($p_0 = 0.3$).

rarely converges to this fair region. Now, let's induce a milder bias by setting $p_0 = 0.3$ and $p_1 = 0.7$ and run the same experiment. The results are illustrated in Fig. 1 (right). As shown in this figure, in the milder bias, the two optimizers behave more similarly, yet RMSProp attains a slightly higher fraction of solutions near $w^*_{pop}$. These observations align with our theoretical findings in Theorem 1. To show that our findings are true for a wider range of $\eta$ and alternative definitions of the fair neighborhood, we refer the reader to Appendix F for additional experiments.

## 3.3 Group fairness analysis

The detailed SDE analysis carried out in Subsection 3.2 highlights the distinct behaviors of SGD and RMSProp concerning group fairness. However, extending this analysis to higher dimensions, particularly through the Fokker-Planck Eq. 10, introduces significant complexity. To address this challenge, we present two theorems that leverage alternative mathematical tools beyond SDE, to study the impact of SGD and RMSProp on group fairness. First, we illustrate how adaptive optimizers like RMSProp suppress subgroup disparities at the level of parameter updates. The second theorem quantifies the consequence, showing that this mechanism translates into provably smaller fairness violations in terms of demographic parity.

**Theorem 2.** *Consider a population that consists of two subgroups with subgroup-specific loss functions $\mathcal{L}_0(w)$ and $\mathcal{L}_1(w)$, sampled with probabilities $p_0$ and $p_1$, respectively. Suppose a stochastic (online) training regime, in which each parameter update is computed from a sample drawn from*

*one of the two subgroups. Suppose the gradients $\nabla\mathcal{L}_0(w_k)$ and $\nabla\mathcal{L}_1(w_k)$ are well-behaved isotropic NGOs, namely $\mathcal{N}(\mu_0, \Theta_0^2 I)$ and $\mathcal{N}(\mu_1, \Theta_1^2 I)$, respectively. Then, the difference in parameter updates between subgroups 0 and 1 under RMSProp has an upper bound given by the corresponding difference under SGD. Consequently, RMSProp offers fairer updates across subgroups.*

*Proof.* For SGD, the difference between the parameter updates across subgroups can be quantified as $||\nabla\mathcal{L}_0 - \nabla\mathcal{L}_1||$. However, as the RMSProp scales the updates using exponentially decaying average, computing the difference is more complex. Let's compute the expected value of $v_k$ defined in Eq. 2:

$$\mathbb{E}[v_k] = \gamma^k v_0 + (1 - \gamma)\Sigma_{i=0}^{k-1}\gamma^i \mathbb{E}[\nabla\mathcal{L}_i^2]. \tag{22}$$

Given that $\mathcal{L}$ (loss function) is a discrete mixture of Gaussians, $\mathbb{E}[\nabla\mathcal{L}_i^2]$ can be calculated as:

$$\mathbb{E}[\nabla\mathcal{L}_i^2] = p_0\left(\mu_0^2 + \Theta_0^2\mathbf{1}\right) + p_1\left(\mu_1^2 + \Theta_1^2\mathbf{1}\right), \tag{23}$$

where $\mathbf{1}$ denotes a vector of all 1s. Substituting the equation above in Eq. 22 yields:

$$\mathbb{E}[v_k] = \gamma^k v_0 + (1 - \gamma^k)[p_0\left(\mu_0^2 + \Theta_0^2\mathbf{1}\right) + p_1\left(\mu_1^2 + \Theta_1^2\mathbf{1}\right)]. \tag{24}$$

The equation above demonstrates that after enough steps, given that $\gamma < 1$, $\mathbb{E}[v_k]$ converges to Eq. 23. Next, let's analyze the variance of the $j$th coordinate of $v_k$, $v_{k_j}$, to see its spread. To this end, we compute the variance of $\nabla\mathcal{L}_{i_j}^2$ using the law of total variance as:

$$var(\nabla\mathcal{L}_{i_j}^2) = \mathbb{E}[var(\nabla\mathcal{L}_{i_j}^2|Z)] + var(\mathbb{E}[\nabla\mathcal{L}_{i_j}^2|Z]), \tag{25}$$

where $Z$ is a random variable denoting the subgroup. Finally, assuming that (a) $\Theta_0^2$ and $\Theta_1^2$ are larger than $\mu_0^2$ and $\mu_1^2$, as shown experimentally in prior work [37], and (b) there is bias in the training set (i.e., $p_0 p_1$ is small), the variance $var(v_{k_j})$ can be computed as $\mathcal{O}((1 - \gamma)(\Theta_{0_j}^4 + \Theta_{1_j}^4))$. For a large $\gamma$, typical in practical training, $var(v_{k_j})$ becomes negligible, and thus $v_k \to E[v_k]$. Using this, we can approximate the difference between the parameter updates across subgroups for RMSProp as $||D(\nabla\mathcal{L}_0 - \nabla\mathcal{L}_1)||$, where $D$ is a diagonal matrix, whose $jj$th element can be computed as: $D_{jj} = \frac{1}{\sqrt{p_0\left(\mu_{0_j}^2 + \Theta_{0_j}^2\right) + p_1\left(\mu_{1_j}^2 + \Theta_{1_j}^2\right) + \epsilon}}$. As $\Theta^2 > \mu^2$ and in stochastic (online) training setup, we can set $\Theta^2 \to 1$, then $D_{jj} < 1$. In other words, the eigenvalues of matrix $D$ is less than 1, thus proving the theorem. $\qquad\square$

Theorem 2 shows that RMSProp, unlike SGD, shrinks disparities between subgroups in the updates. In other words, RMSProp prevents large gradient disparities from dominating the training dynamics. Consequently, the final model is less likely to be pulled toward the subgroup with the larger gradients. Over many iterations, this can help prevent large performance gaps between subgroups. While this does not establish a universal fairness guarantee, it outlines why RMSProp's dynamics yield fairer minima across subgroups. We relax the isotropic noise assumption in Theorem 2 and generalize it to the anisotropic case in Appendix G. The following theorem analyzes fairness through the lens of demographic parity and establishes that, under appropriate conditions in a single iteration, RMSProp's worst-case increase in the gap of demographic parity has an upper-bound no greater than that of SGD.

**Theorem 3.** *Consider a population that consists of two subgroups with subgroup-specific loss functions $\mathcal{L}_0(w)$ and $\mathcal{L}_1(w)$, sampled with probabilities $p_0$ and $p_1$, respectively. Suppose a stochastic (online) training regime in which each parameter update is computed from a sample drawn from one of the two subgroups. Suppose that the gradients $\nabla\mathcal{L}_0(w_k)$ and $\nabla\mathcal{L}_1(w_k)$ are well-behaved isotropic NGOs, namely $\mathcal{N}(\mu_0, \Theta_0^2 I)$ and $\mathcal{N}(\mu_1, \Theta_1^2 I)$, respectively, with $\mu_1 > \mu_0$. Then, in expectation, the worst-case increase in the demographic parity gap after one iteration of RMSProp has an upper-bound no greater than the corresponding increase under SGD.*

*Proof.* See Appendix H. $\qquad\square$

Theorem 3 establishes that the adaptive learning rate of RMSProp, which scales updates based on past squared gradients, helps mitigate demographic parity gaps that can occur during training, whereas SGD does not offer such a mechanism. We relax the isotropic noise assumption in Theorem 3 and generalize the result to the anisotropic case in Appendix I.

## 4 Experiments

**Setup.** We use three public datasets that are widely used in this area: CelebA [31], FairFace [16], and MS-COCO [27]. CelebA and FairFace are datasets commonly used for group fairness analysis, while we also use a general-purpose dataset, MS-COCO, for further completeness. The details about these datasets are presented in Appendix J. The training protocol and backbones used in our experiments are described in Appendix K. We report the performance across different tasks using accuracy and F1. We evaluate group fairness using three widely recognized fairness criteria [38]: equalized odds, equal opportunity, and demographic parity. A predictor is fair under equalized odds definition if the true positive and false positive rates are equal across demographic subgroups. We measure the violation of this criterion as:

$$F_{EOD} = \min_{i,j,q}[min(\frac{p(\hat{y} = c_i|y = c_i, z_j)}{p(\hat{y} = c_i|y = c_i, z_q)}, \frac{p(\hat{y} = c_i|y \neq c_i, z_j)}{p(\hat{y} = c_i|y \neq c_i, z_q)})]. \tag{26}$$

In the equation above, $c_i$ demonstrates class $i$. A classifier satisfies equal opportunity if the true positive rate is equal across demographic subgroups. The related fairness measure is defined as:

$$F_{EOP} = \min_{i,j}(\frac{\sum_{c=1}^{C} TPR_{i,c}}{\sum_{c=1}^{C} TPR_{j,c}}), \tag{27}$$

where $TPR_{i,c}$ denotes the true positive rate for the $i^{\text{th}}$ subgroup on class $c$. Finally, a predictor is said to be fair from a demographic parity point of view if the predicted label distribution is independent of sensitive attributes. The corresponding measure is defined as:

$$F_{DPA} = \min_{i,j,q} \frac{p(\hat{y} = c_i|z_j)}{p(\hat{y} = c_i|z_q)}. \tag{28}$$

For all the above fairness metrics, higher values indicate better (fairer) outcomes.

**Implementation details.** We use PyTorch [42] and up to four Nvidia A100 GPUs. The hyperparameter values are provided in Appendix L, which have been optimized using Weights & Biases [4].

**Results and analysis.** We first calculate the fairness metrics for the ViT backbone across different datasets and different sensitive attributes (gender: G, race: R, and age: A). Fig. 2 presents the results for SGD, RMSProp, and Adam. Considering $F_{EOD}$, Adam consistently outperforms SGD across all scenarios, mostly by substantial margins. For instance, on CelebA dataset with gender as the sensitive attribute, Adam achieves an 8% improvement over SGD. Furthermore, RMSProp outperforms SGD in 4 out of 5 scenarios, and they both achieve zero $F_{EOD}$ for MS-COCO dataset. Given that MS-COCO contains some extremely underrepresented object categories, this result suggests that the disparity captured by $F_{EOD}$ is largely driven by rare categories receiving no correct classifications. The largest gap between RMSProp and SGD is observed on the FairFace dataset with race as the sensitive attribute, where RMSProp outperforms SGD by 9%. Another point to mention is that Adam and RMSProp yield comparable $F_{EOD}$ values, as expected. Considering $F_{EOP}$, both Adam and RMSProp outperform SGD in all 5 scenarios. To shed more light, the largest difference between the two is 6% in the case of FairFace dataset with age as the sensitive attribute. Additionally, unlike SGD, both RMSProp and Adam achieve a near-perfect $F_{EOD}$ in the case of CelebA dataset with gender as the sensitive attribute, indicating their effectiveness in reaching fairer minima.

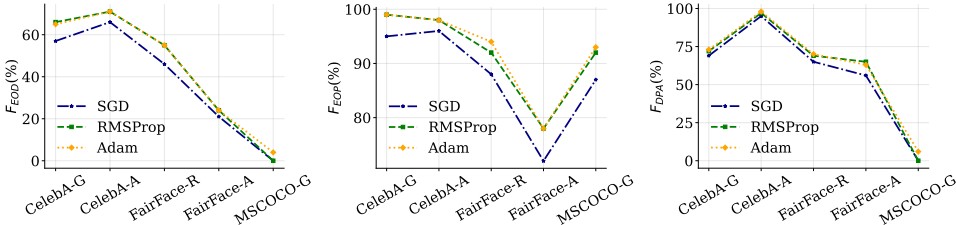

Figure 2: Fairness for ViT across different datasets, attributes (G: gender, A: age, R: race), and metrics.

Regarding $F_{DPA}$, Adam consistently outperforms SGD, while RMSProp surpasses SGD in 4 out of 5 scenarios. Both optimizers achieve zero $F_{DPA}$ on MS-COCO with gender as the sensitive

attribute. Interestingly, the largest discrepancy between RMSProp and SGD occurs on the FairFace dataset with age as the sensitive attribute. This aligns with our theoretical insights from Theorem 1. Specifically, FairFace exhibits extreme imbalance, with a minority-to-all ratio of approximately 0.9%, whereas CelebA with gender as the sensitive attribute has a substantially higher ratio of 42%. These findings highlight that as dataset imbalance intensifies, the gap between adaptive optimizers and SGD becomes more evident. We refer the reader to Appendix M for the results on fairness metrics for other backbones across CelebA, FairFace, and MS-COCO datasets with different sensitive attributes. The results are well-aligned with the results above. To further validate our findings across different variants of optimizers, we conduct additional experiments with SGD w/ momentum [30], AdamW [32], and AdaBound [33] optimizers. The results are provided in Appendix N.

Beyond tuning the learning rate, we further investigate the influence of additional hyperparameters on fairness outcomes. Specifically, we extend the hyperparameter tuning using Bayesian Optimization over the learning rate, decay rate, and batch size. This comprehensive search is performed using the ViT backbone on the CelebA dataset, considering gender and age as sensitive attributes. The results are reported in Table 1. As shown, adaptive optimizers like Adam still achieve better fairness compared to SGD. This confirm that the fairness trends observed in our main study remain consistent even when multiple hyperparameters are jointly tuned.

Table 1: Fairness comparison across optimizers with tuning learning rate, decay rate, and batch size.

|  | Gender | | | Age | | |
|---|---|---|---|---|---|---|
|  | Adam | RMSProp | SGD | Adam | RMSProp | SGD |
| $F_{EOD}$ | 65.21 | 65.18 | 62.66 | 72.34 | 71.99 | 68.40 |
| $F_{EOP}$ | 99.90 | 99.91 | 96.60 | 99.10 | 99.10 | 97.77 |
| $F_{DPA}$ | 73.50 | 73.68 | 60.80 | 98.20 | 98.20 | 95.40 |

To demonstrate that our training is successful and the models are well-behaved, we evaluate the classification performance of ViT on the three datasets for facial expression recognition, multi-label classification, and gender classification, respectively. The results are presented in Table 2. The performance of the other models are presented in Appendix O. Here, we can confirm that

Table 2: Comparison of accuracy and F1 across different optimizers and datasets.

| Dataset | Accuracy | | | F1 score | | |
|---|---|---|---|---|---|---|
|  | SGD | RMSProp | Adam | SGD | RMSProp | Adam |
| CelebA | 91.23 | 91.54 | 92.08 | 92.12 | 91.17 | 92.09 |
| MS-COCO | 89.62 | 89.71 | 90.03 | 68.35 | 71.03 | 74.10 |
| FairFace | 89.41 | 91.37 | 92.20 | 91.13 | 92.07 | 92.17 |

in most cases, all three optimizers exhibit comparable accuracy and F1 scores across all tasks. For instance, on the CelebA dataset, SGD and RMSProp achieve an accuracy of 91%, while Adam attains 92%. This trend is consistent with previous findings in the prior work [34]. As evident, the improved fairness observed with adaptive gradient algorithms cannot be solely attributed to their performance. For instance, on the CelebA dataset, the F1 score for SGD is higher than that of RMSProp, whereas on FairFace, RMSProp achieves a higher F1 score than SGD. However, in both cases, RMSProp demonstrates better fairness performance, as shown in Fig. 2. This further supports our theoretical conclusion that RMSProp is fairer than SGD, regardless of overall performance.

To rigorously assess the significance of fairness improvement of Adam and RMSProp over SGD, we conduct the following experiment. We train ViT on CelebA dataset 10 times using SGD, RMSProp, and Adam, while recording fairness metrics for both gender and age attributes in each run. To evaluate statistical significance, we perform a paired Wilcoxon signed-rank test [3] on the fairness metrics and report the corresponding p-values in Table 3. As shown in the table, most scenarios yield *p*-values in the order of 0.001, indicating a statistically significant improvement in

Table 3: *p*-values from the Wilcoxon test comparing SGD vs. RMSProp and SGD vs. Adam optimizers on the CelebA dataset for gender and age attributes.

| Metric | Gender | | Age | |
|---|---|---|---|---|
|  | SGD-RMSProp | SGD-Adam | SGD-RMSProp | SGD-Adam |
| $F_{EOD}$ | $1 \times 10^{-3}$ | $1 \times 10^{-3}$ | $1 \times 10^{-3}$ | $1 \times 10^{-3}$ |
| $F_{EOP}$ | $1 \times 10^{-3}$ | $1 \times 10^{-3}$ | $1 \times 10^{-3}$ | $5 \times 10^{-3}$ |
| $F_{DPA}$ | $2 \times 10^{-3}$ | $1 \times 10^{-3}$ | $7 \times 10^{-3}$ | $3 \times 10^{-3}$ |

fairness for adaptive gradient optimizers compared to SGD. The worst *p*-value in the table corresponds to SGD vs. RMSProp on CelebA with age as the sensitive attribute. However, even this value remains statistically significant under the conventional threshold of 0.05 [14, 40].

Theorem 1 provides theoretical insight that under the demographic parity fairness criterion, RMSProp is more likely to converge to a fairer minimum than SGD in highly imbalanced datasets. Moreover, it predicts that as the dataset becomes more balanced, the fairness advantage of RMSProp over SGD

Table 4: Comparison of fairness metrics across optimizers, with and without fairness-enhancing methods. Lower values indicate better fairness.

| Dataset | Gap in Equal Opportunity | | | Gap in Equalized Odds | | | Gap in Demographic Parity | | |
| --- | --- | --- | --- | --- | --- | --- | --- | --- | --- |
| | Adam | RMSProp | SGD | Adam | RMSProp | SGD | Adam | RMSProp | SGD |
| *With fairness-enhancing* | | | | | | | | | |
| ProPublica COMPAS | 0.45 | 0.48 | 0.71 | 2.79 | 2.78 | 2.90 | 0.86 | 0.86 | 2.60 |
| AdultCensus | 3.15 | 3.16 | 3.38 | 2.39 | 2.41 | 4.28 | 7.00 | 6.80 | 11.79 |
| *Without fairness-enhancing* | | | | | | | | | |
| ProPublica COMPAS | 13.99 | 13.90 | 15.19 | 13.99 | 13.95 | 14.98 | 11.49 | 11.45 | 11.80 |
| AdultCensus | 21.04 | 20.91 | 21.29 | 20.90 | 20.91 | 21.14 | 12.19 | 12.23 | 12.42 |

gradually diminishes. To empirically validate this behavior, we conduct an experiment where we randomly downsample the 'male' subgroup in the CelebA dataset from its native male-to-all ratio of 42% down to 22% and 2%. We then train a model on each downsampled version using both RMSProp and SGD. We repeat this process five times while recording the values of $F_{DPA}$ and $F_{EOD}$. Fig. 3 shows the results, illustrating the absolute difference in $F_{DPA}$ and $F_{EOD}$ between SGD and RMSProp across different male-to-all ratios. As expected, the gap in demographic parity is maximum at the most severe imbalance level (2% male-to-all ratio). As the male-to-all ratio increases (indicating less imbalance), the absolute difference between the two optimizers gradually decreases, a trend that also holds for equalized odds. This empirical finding strongly aligns with the theoretical predictions in Theorem 1.

Theorems 2 and 3 are general with respect to the loss function, which suggests their fairness benefits should be complementary to existing fairness-enhancing methods that add regularization terms to the primary loss. To validate this hypothesis, we conduct an experiment where we test the impact of optimizers in addition to the fairness-enhancing method [47] on two tabular datasets (ProPublica COMPAS [2] and AdultCensus [19]) with gender as the sensitive attribute. The results are presented in Table 4 for SGD, Adam, and RMSProp. Following [47], we report the average gap in equal opportunity, gap in equalized odds, and gap in demographic parity (lower is better) over 10 runs. Regarding the fairness enhancing method, across all metrics on both datasets, Adam and RMSProp achieve substantially lower (better) fairness gaps than SGD. This

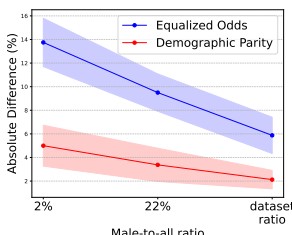

Figure 3: Difference of RMSProp and SGD's fairness in different male-to-all ratios for CelebA dataset with gender as sensitive attribute.

experiment demonstrates that the fairness benefits of adaptive optimizers are not limited to standard training but are complementary to and can amplify the effectiveness of existing fairness interventions. Additionally, Table 4 shows that in the absence of any fairness-enhancing method, adaptive optimizers yield better fairness outcomes than SGD. This finding further indicates that the fairness advantage of adaptive optimizers extends beyond computer vision applications.

## 5    Conclusion

This paper provided both theoretical and empirical evidence that the choice of optimizer can significantly affect the group fairness of resulting deep learning models. Through establishing an analytically tractable setup and analyzing it via stochastic differential equations, we showed that SGD and RMSProp can converge to different minima with regards to fairness, with RMSProp more frequently reaching fairer optima. Next, we provided two theorems that showed 1) adaptive gradient methods (e.g., RMSProp) shrink subgroup-specific gradients more effectively, thereby reducing unfair updates compared to their SGD counterparts; 2) In a single optimization step, the worst case demographic disparity of RMSProp has the upper bound given by that of SGD. We validated these theoretical findings on three diverse datasets, namely CelebA, FairFace, and MS-COCO, and on multiple tasks. Across a range of group fairness criteria, i.e, equalized odds, equal opportunity, and demographic parity, adaptive optimizers consistently led to fairer solutions. We hope our work will encourage broader adoption of adaptive optimizers in fairness-critical domains, and we envision this research as a step toward developing more equitable deep learning models.

## Acknowledgment

This work was partially funded by the Natural Sciences and Engineering Research Council of Canada (NSERC), and was enabled in part by the support provided by the Digital Research Alliance of Canada.

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

## Appendix

## A Itô's Lemma

Itô's lemma extends the chain rule of classical calculus to stochastic calculus, thereby accommodating functions that depend on Brownian motion $B(t)$. Suppose that the following stochastic differential equation (SDE) is given:

$$dX(t) = M(t, X(t))dt + N(t, X(t))dB(t). \tag{29}$$

In the equation above, X(t) is a stochastic process, M() and N() are the functions of $X(t)$ and $t$. Let's define stochastic process $Y(t)$ by:

$$Y(t) = Q(t, X(t)), \tag{30}$$

where, $Q()$ is a function of stochastic process $X(t)$ and $t$. Itô's lemma states that we can compute $dY(t)$ by using the following:

$$dY(t) = \left( \frac{d}{dt}Q(t, X) + \frac{d}{dx}Q(t, X) \cdot M(t, X(t)) + 0.5\frac{d^2}{dx^2}Q(t, X)N^2(t, X(t)) \right) dt +$$
$$\left( \frac{d}{dx}Q(t, X) \cdot N(t, X(t)) \right) dB(t). \tag{31}$$

In the above equation, as the term $0.5\frac{d^2}{dx^2}Q(t, X)N^2(t, X(t))$ does not appear in classic calculus, it is often referred to as Itô's correction term.

## B Fokker-Planck equation

In this section, for completeness, we provide the derivation of fokker-planck equation from a given SDE. This is a well-known equation in the analysis of SDEs. Suppose the following SDE is given:

$$dX(t) = M(t, X(t))dt + N(t, X(t))dB(t). \tag{32}$$

Also, suppose that $Q(x)$ is a function which is 0 outside of a bounded interval. For simplicity, we use $X$ instead of $X(t)$. By Itô's lemma, we have:

$$dQ(X) = \left( M(t, X)Q'(X) + 0.5N^2(t, X)Q''(X) \right) dt + N(t, X)Q'(X)dB(t). \tag{33}$$

Let's integrate the equation above from 0 to $t$ and then take the expected value from both sides:

$$\mathbb{E}[Q(X) - Q(0)] = \int_0^t \mathbb{E}[\left( M(\tau, X)Q'(X) + 0.5N^2(\tau, X)Q''(X) \right)]d\tau +$$
$$\int_0^t \mathbb{E}[N(\tau, X)Q'(X)dB(\tau)]. \tag{34}$$

Suppose $X$ is independent of $B$, then as $\mathbb{E}[dB(\tau)] = 0$, the second integral above is zero. Next, let's take the derivative with regards to $t$ from both sides:

$$\frac{d}{dt}\mathbb{E}[Q(X)] = \mathbb{E}[\left( M(t, X)Q'(X) + 0.5N^2(t, X)Q''(X) \right)] \tag{35}$$

To make the distribution of $X$ appear in the equations, let's use the definition of $\mathbb{E}[.]$ in the equation above:

$$\int_{-\infty}^{+\infty} \frac{\partial}{\partial t}p(t, x)Q(x) = \int_{-\infty}^{+\infty} p(t, x)M(t, x)Q'(x)dx + \int_{-\infty}^{+\infty} p(t, x)0.5N^2(t, x)Q''(x)dx \tag{36}$$

In the equation above, $p(t, x)$ is the distribution of stochastic process $X$. By applying integration by part to the integrals in the Eq. 36, and re-arranging the terms, we get:

$$\int_{-\infty}^{+\infty} \left( \frac{\partial}{\partial t}p(t, x) + \frac{\partial}{\partial x}(p(t, x)M(t, x)) - 0.5\frac{\partial^2}{\partial x^2}(p(t, x)N^2(t, x)) \right) Q(x)dx = 0 \tag{37}$$

As the equation above holds for all $Q(x)$, then the other factor must equal to zero. Thus, we have:

$$\frac{\partial}{\partial t}p(t, x) + \frac{\partial}{\partial x}(p(t, x)M(t, x)) - 0.5\frac{\partial^2}{\partial x^2}(p(t, x)N^2(t, x)) = 0 \tag{38}$$

The equation above is known as Fokker-Planck equation.

## C  Well-behaved NGOS

Noisy Gradient Oracle with Scale Parameter (NGOS) models stochastic gradients as $g(w) = \nabla \mathcal{L}(w) + \Theta z$, where $\Theta$ is a noise scale parameter, and $z$ is a random variable that follows a distribution with covariance matrix $\Sigma(w)$ and a mean of zero. An NGOS is well-behaved if the conditions below are satisfied [37]:

*1.* $\nabla \mathcal{L}(w)$ is Lipschitz.

*2.* Square root of covariance matrix is Lipschitz and bounded.

*3.* Both $\nabla \mathcal{L}(w)$ and $\Sigma(w)$ are differentiable and their partial derivatives up to 4-th order have polynomial growth.

*4.* Noise distribution in the NGOS must have low skewness. In other words, there must exist a function of polynomial growth such that $|\mathbb{E}[z^3]|$ is equal or less than that function divided by $\Theta$.

*5.* Noise distribution must have bounded moments. In other words, there must exist a constant $\kappa$ such that $\mathbb{E}[||z||^{2k}]^{\frac{1}{2k}} \leq \kappa(1 + ||w||)$.

Gaussian noise is a well-behaved NGOS. However, the noise in NGOS is not restricted to Gaussian. Particularly, if the noise distribution is not heavy-tailed, then it satisfies requirements 4 and 5. Whether or not the noise distribution in real-world applications is heavy-tailed is still an open problem. Nevertheless, several experimental results denote that the conditions described above are not too strong [37].

## D  Approximation Quality of SDE for SGD

The dynamics of SGD can be approximated by the following SDE [25]:

$$dW_t = -\nabla \mathcal{L}(W_t)dt + (\eta \Sigma(W_t))^{1/2}dB(t), \tag{39}$$

where $W_t$ represents an approximation of $w_k$ at discrete time steps $k\,\eta$ and $\Sigma$ is the covariance matrix of the underlying NGOS. The noises that make the individual paths for the above SDE and the SGD are independent processes. Hence, SGD and its SDE approximation do not share the same sample paths; rather, they are close to each other in "distribution" or "weak sense" [25]. To formalize this mathematically, suppose $\Xi$ be the set of functions, $f(x)$, of polynomial growth, i.e. $\exists a, b \in \mathbb{R}^+$ such that $|f(x)| < a(1 + |x|^b)$. Then, SDE defined in Eq. 39 is the order $r$ weak approximation of SGD if for each function $f(x)$ in $\Xi$, there exists a constant $C$ such that for all $k = 1 \ldots N$, $|\mathbb{E}[f(W_{k\eta})] - \mathbb{E}[f(w_k)]| < C\eta^r$. This is a well-known result in analyzing optimization algorithms using SDEs, first time proposed in [25]. Similar theorems for RMSProp and Adam are available in [37].

## E  Proof of Lemma 1

The demographic parity criterion for fairness between two subgroups is defined as:

$$P(\hat{y}|z = 0) = P(\hat{y}|z = 1), \tag{40}$$

where $z$ denotes the subgroup, and $\hat{y}$ represents the output of a model. The gap in demographic parity serves as a fairness measure [38]. A common approach to quantifying this gap is through the zero-one loss difference [24]: $|[\ell_{0/1}(w)|z = 0] - [\ell_{0/1}(w)|z = 1]|$. As established in prior work [24], the zero-one loss can be approximated by the training loss, leading to the following fairness measure $\mathcal{F}(w) = |[\mathcal{L}(w)|z = 0] - [\mathcal{L}(w)|z = 1]|$. Substituting the given loss functions $\mathcal{L}_0(w)$ and $\mathcal{L}_1(w)$, we obtain $\mathcal{F}(w) = 2|w|$. Since $\mathcal{F}(w)$ achieves its minimum at $w = 0$, this proves that $w = 0$ is the fairest minimizer.

## F  Additional Results for Theorem 1

In this section, we extend our empirical analysis of Theorem 1 by evaluating our findings with broader choices of learning rates and alternative definitions of the fair neighborhood.

Table 5: Hyperparameter configurations for the experimental scenarios.

| Configuration | $\eta_{RMSProp}$ | $\eta_{SGD}$ | Fairness Threshold |
|---|---|---|---|
| 1 | 0.01 | 0.1 | 0.2 |
| 2 | 0.1 | 0.2 | 0.2 |
| 3 | 0.01 | 0.1 | 0.1 |
| 4 | 0.01 | 0.1 | 0.4 |
| 5 | 0.1 | 0.2 | 0.1 |
| 6 | 0.1 | 0.2 | 0.4 |

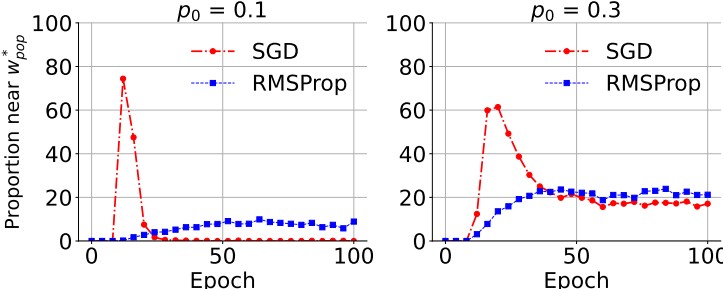

Figure 4: Convergence rates of RMSProp ($\eta = 0.01$) and SGD ($\eta = 0.1$) to the fair neighbourhood, defined by threshold of $0.2$.

Theorem 1 establishes that for subgroup sampling probabilities $p_0, p_1 \in (0, 1)$ with $p_0 + p_1 = 1$, and loss functions $\mathcal{L}_0(w) = \frac{1}{2}(w - 1)^2$ and $\mathcal{L}_1(w) = \frac{1}{2}(w + 1)^2$, the optimization of the empirical loss function:

$$\mathcal{L}_{emp}(w) \;=\; \frac{1}{N} \sum_{r \in \Omega} \mathcal{L}_{q_r}(w), \tag{41}$$

where each sample $q_r \in 0, 1$ is drawn i.i.d. with probability $p_0$ for subgroup 0 or $p_1$ for subgroup 1, leads to an increased likelihood of RMSProp converging to the fair optimum $w_{pop}^* = 0$ as the imbalance between $p_0$ and $p_1$ grows. To empirically validate these theoretical insights, we conducted simulations using a fixed learning rate of $\eta = 0.1$, training for 1000 independent runs per experiment. The fraction of runs converging to the neighborhood of the fair optimum, defined as $|w - w_{pop}^*| < 0.2$, was recorded at each epoch. We call $0.2$ the fairness threshold. To assess the robustness of our findings, we extend our analysis by examining a broader range of $\eta$ values and explore more restrictive and relaxed fairness thresholds. Table 5 summarizes the hyperparameter configurations used in our experiments. Fig.s 4 to 9 illustrate the convergence behavior of RMSProp and SGD under the specified configurations in the table.

The figures reveal a consistent trend in the convergence behavior of RMSProp and SGD. In the severely biased setting ($p_0 = 0.1$), RMSProp exhibits a higher likelihood of converging to the fair neighborhood, regardless of the learning rate. As the bias decreases ($p_0 = 0.3$), the behavior of RMSProp and SGD becomes more similar. These experimental findings are in line with theoretical findings in Theorem 1. Additionally, a more restrictive fairness threshold reduces the probability of convergence to the fair neighborhood, whereas a relaxed threshold increases it. This is expected, as stricter definitions impose tighter constraints on acceptable fair solutions, while relaxed criteria naturally allow a higher fraction of runs to qualify as fair.

## G  Extension of Theorem 2 to Anisotropic Noise

**Theorem 4.** *Consider a population that consists of two subgroups with subgroup-specific loss functions $\mathcal{L}_0(w)$ and $\mathcal{L}_1(w)$, sampled with probabilities $p_0$ and $p_1$, respectively. Suppose a stochastic (online) training regime, in which each parameter update is computed from a sample drawn from one of the two subgroups. Suppose the gradients $\nabla \mathcal{L}_0(w_k)$ and $\nabla \mathcal{L}_1(w_k)$ are well-behaved anisotropic NGOs, namely $\mathcal{N}(\mu_0, \Sigma_0)$ and $\mathcal{N}(\mu_1, \Sigma_1)$, respectively. Then, the difference in parameter updates*

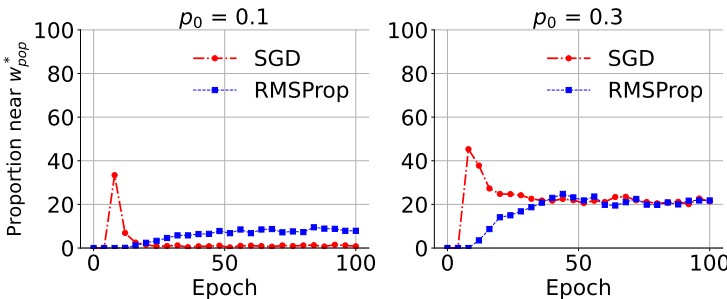

Figure 5: Convergence rates of RMSProp ($\eta = 0.1$) and SGD ($\eta = 0.2$) to the fair neighbourhood, defined by threshold of $0.2$.

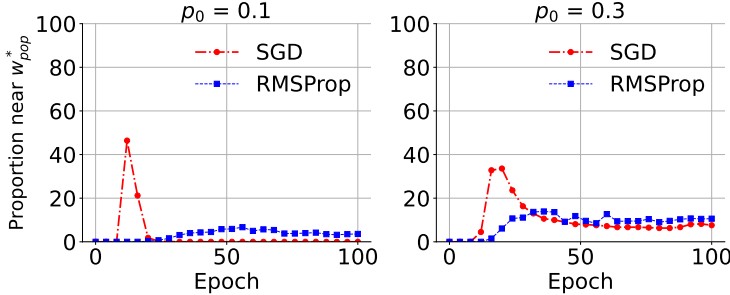

Figure 6: Convergence rates of RMSProp ($\eta = 0.01$) and SGD ($\eta = 0.1$) to the fair neighbourhood, defined by threshold of $0.1$.

*between subgroups 0 and 1 under RMSProp has an upper bound given by the corresponding difference under SGD. Consequently, RMSProp offers fairer updates across subgroups.*

*Proof.* The proof is similar to Theorem 2. For SGD, the difference between the parameter updates across subgroups is $||\nabla\mathcal{L}_0 - \nabla\mathcal{L}_1||$. Regarding RMSProp, let's compute the expected value and the variance of $v_k$:

$$\mathbb{E}[v_k] = \gamma^k v_0 + (1 - \gamma^k)[p_0\left(\mu_0^2 + diag(\Sigma_0)\right) + p_1\left(\mu_1^2 + diag(\Sigma_1)\right)]. \tag{42}$$

Where $diag(\Sigma_0)$ and $diag(\Sigma_1)$ are the vectors with the diagonal elements of $\Sigma_0$ and $\Sigma_1$, respectively. Similar to the the isotropic case, we can show that the variance of $v_k$ is negligible and thus $v_k \to E[v_k]$. Hence, the difference between parameter updates are $||D(\nabla\mathcal{L}_0 - \nabla\mathcal{L}_1)||$, where $D$ is a diagonal matrix, whose $jj$th element can be computed as:

$$D_{jj} = \frac{1}{\sqrt{p_0\left(\mu_{0_j}^2 + \sigma_{0_j}^2\right) + p_1\left(\mu_{1_j}^2 + \sigma_{1_j}^2\right) + \epsilon}}. \tag{43}$$

Prior work [37] has experimentally shown that $\sigma^2 > \mu^2$. Additionally, in stochastic (online) training setup: $\Theta^2 \to 1$. Thus, $D_{jj} < 1$. This completes the proof. $\square$

# H   Proof of Theorem 3

*Proof.* As described in the proof of Lemma 1, the following can be used to approximate the gap in demographic parity: $\mathcal{F}(w) = |[\mathcal{L}(w)|z = 0] - [\mathcal{L}(w)|z = 1]|$. Let's define the function $\Psi(w_k) = \mathcal{L}_0(w_k) - \mathcal{L}_1(w_k)$. Also, define $\varphi(w_k) = \frac{L_0(w_k) - L_1(w_k)}{|L_0(w_k) - L_1(w_k)|}$. Let's expand $\mathcal{F}(w_k)$ around $w_k$ using Taylor series:

$$\mathcal{F}(w_k + \delta) - \mathcal{F}(w_k) = \nabla\mathcal{F}(w_k)^t\delta. \tag{44}$$

We can compute $\nabla\mathcal{F}(w_k)$ as:

$$\nabla\mathcal{F}(w_k) = \varphi(w_k) \cdot (\nabla\Psi(w_k)). \tag{45}$$

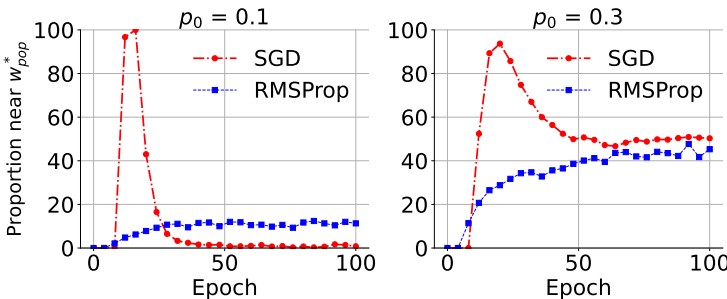

Figure 7: Convergence rates of RMSProp ($\eta = 0.01$) and SGD ($\eta = 0.1$) to the fair neighbourhood, defined by threshold of $0.4$.

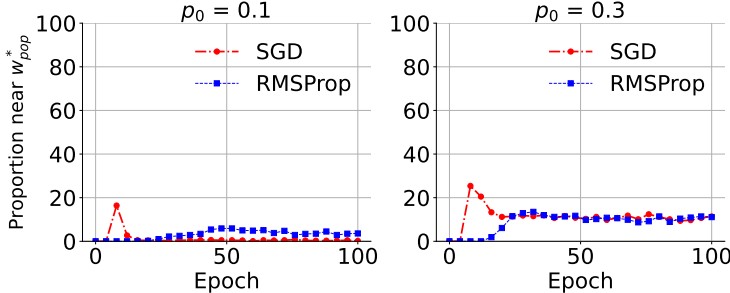

Figure 8: Convergence rates of RMSProp ($\eta = 0.1$) and SGD ($\eta = 0.2$) to the fair neighbourhood, defined by threshold of $0.1$.

If in Eq. 44 we set $\delta - \eta \nabla \mathcal{L}(w_k)$, given the updates of SGD in Eq. 1, we can re-write Eq. 44 as:

$$\mathcal{F}(w_{k+1_{sgd}}) - \mathcal{F}(w_k) = -\eta \varphi(w_k) \nabla \Psi(w_k)^t \nabla \mathcal{L}, \tag{46}$$

where $\mathcal{F}(w_{k+1_{sgd}})$ is the value of function $\mathcal{F}()$ after one iteration using SGD. Using the proof of Theorem 2, we can perform a similar operation for RMSProp:

$$\mathcal{F}(w_{k+1_{rms}}) - \mathcal{F}(w_k) = -\eta \varphi(w_k) \nabla \Psi(w_k)^t D \nabla \mathcal{L}. \tag{47}$$

In the equation above, $\mathcal{F}(w_{k+1_{rms}})$ is the value of $\mathcal{F}()$ after one iteration using RMSProp, and matrix $D$ is a diagonal matrix defined in the proof of Theorem 2. As $\mu_0 < \mu_1$, then the worst case increase in the gap of demographic parity for SGD occurs when $\nabla \Psi(w_k)$ aligns with $\nabla \mathcal{L}$. Mathematically, the increase can be modeled as:

$$|\mathcal{F}(w_{k+1_{sgd}}) - \mathcal{F}(w_k)| = \eta |\varphi(w_k) \nabla \Psi(w_k)^t \nabla \mathcal{L}| \leq \eta \|\nabla \Psi(w_k)\| \cdot \|\nabla \mathcal{L}\|. \tag{48}$$

Similarity, for RMSProp, we have:

$$\begin{aligned} |\mathcal{F}(w_{k+1_{rms}}) - \mathcal{F}(w_k)| = \eta |\varphi(w_k) \nabla \Psi(w_k)^t D \nabla \mathcal{L}| &\leq \eta \|D \nabla \Psi(w_k)\| \cdot \|\nabla \mathcal{L}\| \\ &< \eta \|\nabla \Psi(w_k)\| \cdot \|\nabla \mathcal{L}\|. \end{aligned} \tag{49}$$

In the equation above, as diagonal elements of $D$ are strictly less than 1, we have a strict inequality. Comparing Eqs. 48 and 49, we conclude the proof. $\qquad \square$

# I  Extension of Theorem 3 to Anisotropic Noise

**Theorem 5.** *Consider a population that consists of two subgroups with subgroup-specific loss functions $\mathcal{L}_0(w)$ and $\mathcal{L}_1(w)$, sampled with probabilities $p_0$ and $p_1$, respectively. Suppose a stochastic (online) training regime in which each parameter update is computed from a sample drawn from one of the two subgroups. Suppose that the gradients $\nabla \mathcal{L}_0(w_k)$ and $\nabla \mathcal{L}_1(w_k)$ are well-behaved anisotropic NGOs, namely $\mathcal{N}(\mu_0, \Sigma_0)$ and $\mathcal{N}(\mu_1, \Sigma_1)$, respectively, with $\mu_1 > \mu_0$. Then, in expectation, the worst-case increase in the demographic parity gap after one iteration of RMSProp has an upper-bound no greater than the corresponding increase under SGD.*

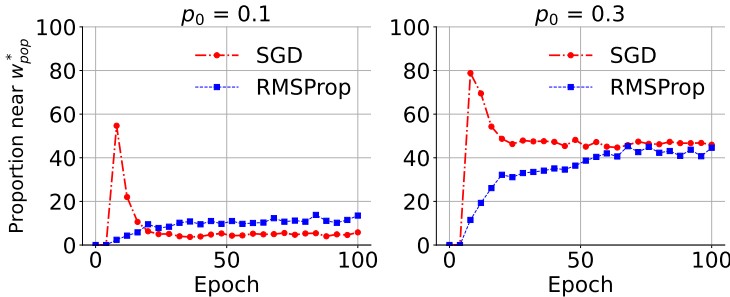

Figure 9: Convergence rates of RMSProp ($\eta = 0.1$) and SGD ($\eta = 0.2$) to the fair neighbourhood, defined by threshold of $0.4$.

*Proof.* The proof follows directly as the proof of this Theorem relies on the conclusion of the extension of the Theorem 2, which is already proved in Theorem 4. □

## J Datasets

**CelebA** [31] is a large-scale and real-world dataset, which include 202,599 facial images from over 10,000 individuals. Each image is annotated with 40 distinct attributes, and the dataset is officially partitioned into training, validation, and test sets. In this study, we use gender (male vs. female) and age (young vs. old) as sensitive attributes, while performing facial expression recognition based on the "happy" attribute. Notably, the dataset exhibits an imbalance in demographic distribution, with the probability of an image depicting a male being 0.42 and the probability of depicting a young individual being 0.77. This skewed distribution indicates the presence of bias in the training set.

**FairFace** [16] is a large-scale face attribute dataset, mainly designed to mitigate racial bias in facial analysis tasks. It includes 108,501 facial images sourced primarily from the YFCC-100M Flickr dataset [53]. Each image is annotated for gender, age (9 sub-groups), and race (7 sub-groups). In this study, we leverage FairFace for gender classification, treating age and race as sensitive attributes. Given its broad demographic representation, FairFace serves as a robust benchmark in fairness related tasks.

**MS-COCO** [27] is a large-scale, real-world benchmark dataset widely used for multi-label classification, object detection, and image captioning. It consists of over 330,000 images, including more than 200,000 labeled images spanning 80 object categories. In this study, we leverage MS-COCO for multi-label classification. Notably, the dataset does not provide explicit gender annotations. To infer gender labels, we follow a protocol established in previous works [57], utilizing the five associated captions per image. Specifically, we retain images where at least one caption contains either the term "man" or "woman". Additionally, we remove object categories that are not strongly associated with humans. Human-associated objects are defined as those appearing at least 100 times alongside either "man" or "woman" in the captions, resulting in a reduced set of 75 object categories. Fig. 10 provides sample captions and their corresponding inferred gender labels. It is worth mentioning that the probability of an image depicting a male is 0.71.

## K Training Protocol

For facial expression recognition, we employ RetinaFace [8] to detect and crop faces from images. The images in the FairFace dataset are already cropped by default. As part of the preprocessing pipeline for all tasks, we normalize the images and resize them to $224 \times 224$. Data augmentation consists of random horizontal flipping and random rotation. Facial expression recognition and gender classification tasks are trained using the cross-entropy loss $\mathcal{L}_{ce}$, defined as:

$$\mathcal{L}_{ce} = -\frac{1}{N} \sum_{j=1}^{N} \left[ y_j \log(\hat{y}_j) + (1 - y_j) \log(1 - \hat{y}_j) \right]. \tag{50}$$

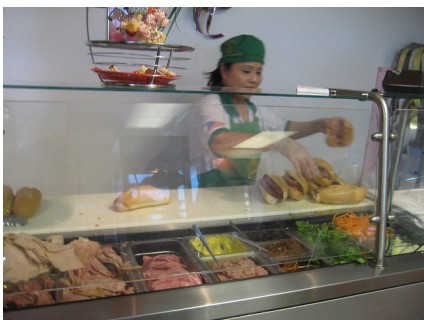
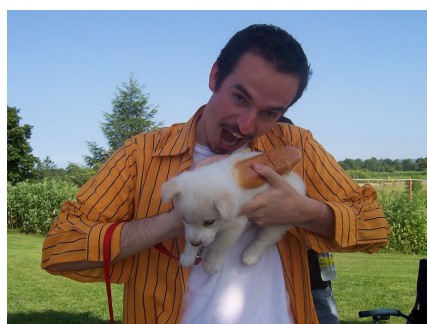

(a) **1.** a woman prepares several sub sandwiches at a deli counter.
**2.** a person behind a display glass preparing food
**3.** a lady behind a sneeze guard making sub sandwiches.
**4.** a woman behind a deli counter making sub sandwiches.
**5.** there is a woman that is making sandwiches at a deli

(b) **1.** a man in a yellow shirt holding a white dog and making a face
**2.** a man holding a hot dog bun beside his puppy.
**3.** a man putting a hot dog bun on a puppy and pretending to eat
**4.** a man holding a white puppy and a red leash.
**5.** a man who is pretending to bite a puppy.

Figure 10: Two example images from MS-COCO dataset along with their captions. The inferred gender labels from the captions for the left and right images are "woman" and "man", respectively.

For the multi-label classification task, given the class imbalance in MS-COCO, we employ the focal loss [28] to mitigate overfitting to majority classes. It is formulated as:

$$\mathcal{L}_{focal} = -\frac{1}{N}\sum_{j=1}^{N}[\xi y_j(1-\hat{y}_j)^{\upsilon}\log(\hat{y}_j) + (1-\xi)(1-y_j)\hat{y}_j^{\upsilon}\log(1-\hat{y}_j)], \qquad (51)$$

where $\xi$ is the balancing factor and $\upsilon$ is the focusing parameter. To optimize the learning rate during training, we leverage Bayesian optimization [50] combined with the Hyperband algorithm [23]. The best-performing models are trained twice, and we report the average performance across these runs.

We experiment with multiple backbone architectures, including ResNet-18 and ResNet-50 [11], VGG-16 [49], and a recently proposed vision transformer (tiny ViT)[54]. These architectures have been widely used in studies analyzing the impact of optimization methods on different learning paradigms[34], as well as in fairness-focused works [55]. ResNet models leverage residual connections to facilitate deep representation learning, while VGG-16 provides a structured CNN design with uniform filter sizes. The tiny ViT, a transformer-based model, processes images as patch embeddings, and showed state-of-the-art performance [54]. This diverse selection allows us to investigate fairness effects across both convolutional and transformer-based architectures.

## L  Hyperparameters

We employ Bayesian Optimization [50] with Hyperband [23] algorithm to systematically tune the learning rate across different optimizers. Hyperband efficiently allocates training resources to more promising setups and progressively eliminate underperforming ones. It evaluates training at specific checkpoints, called "brackets". We set Hyperband brackets to [10, 20, 40], allowing enough time for each run to demonstrate its potential. Additionally, the learning rate is sampled using a log-uniform distribution within the following search spaces:

- SGD: $[10^{-3}, 10^{-1}]$
- RMSProp: $[10^{-5}, 10^{-2}]$
- Adam: $[10^{-5}, 10^{-2}]$

All models are trained for maximum of 60 epochs with a batch size of 64 and weight decay of $10^{-4}$. We use Pytorch default values for optimizer specific hyperparameters:

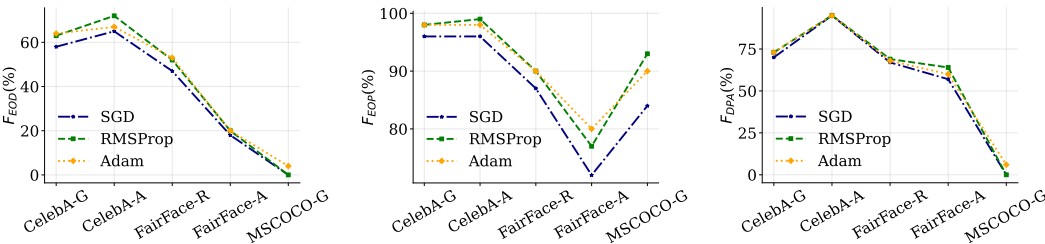

Figure 11: Fairness metrics for ResNet-18 across different datasets and sensitive attributes.

- RMSProp: $\gamma = 0.9$

- Adam: $\beta_1 = 0.9$, $\beta_2 = 0.999$

Additionally, we use "Reduce-on-Plateau" scheduler of Pytorch to decrease the learning rate during the training. if no improvement is observed for 5 epochs, then the learning rate is multiplied by 0.5.

## M    Additional Fairness Results across Different Backbones and Datasets

In this section, we present fairness metrics for SGD, RMSProp, and Adam across ResNet-18, ResNet-50, and VGG-16 backbones, evaluated on multiple datasets with different sensitive attributes.

**ResNet-18:** Fig. 11 illustrates fairness metrics for the ResNet-18 backbone. Considering $F_{EOD}$, both Adam and RMSProp consistently outperform SGD across all datasets and sensitive attributes. The only exception is the MS-COCO dataset, where both RMSProp and SGD yield an $F_{EOD}$ of zero. This is due to MS-COCO's 75 object classes, some of which are highly underrepresented, leading to an effective fairness score of zero. The largest difference in $F_{EOD}$ is observed on the CelebA dataset with age as the sensitive attribute, where RMSProp significantly surpasses SGD. Adam and RMSProp exhibit comparable performance, as expected.

For the $F_{EOP}$ metric, both RMSProp and Adam outperform SGD across all datasets. For example, in the FairFace dataset with age as the sensitive attribute, Adam achieves an $F_{EOP}$ of 80%, compared to 72% with SGD. Regarding $F_{DPA}$, Adam surpasses SGD in four out of five cases. The only exception is CelebA with age as the sensitive attribute, where both optimizers yield similar $F_{DPA}$. This aligns with Theorem 1, as CelebA is not sufficiently biased for adaptive gradient methods to demonstrate a significant fairness advantage.

**ResNet-50:** Fig. 12 presents results for the ResNet-50 backbone. For $F_{EOD}$, Adam consistently outperforms SGD, while RMSProp surpasses SGD in four out of five cases, with MS-COCO again being the exception. The difference between Adam and SGD reaches approximately 15% in CelebA when using age as the sensitive attribute. Similar trends emerge for $F_{EOP}$, where both Adam and RMSProp outperform SGD across all datasets. Finally, for $F_{DPA}$, more biased datasets, such as FairFace with age as the sensitive attribute, show a higher gap between adaptive gradient methods and SGD.

**VGG-16:** Fig. 13 shows fairness metrics for the VGG-16 backbone. As with the previous architectures, Adam outperforms SGD in all cases for $F_{EOD}$. For $F_{EOP}$, both Adam and RMSProp demonstrate superior fairness compared to SGD. In FairFace with age as the sensitive attribute, the difference between Adam and SGD reaches approximately 10%. For $F_{DPA}$, the most significant difference is again observed in FairFace with age as the sensitive attribute, which reinforces our theoretical findings.

Across all three fairness metrics, we observe consistent trends across multiple architectures, datasets, and problems, including gender classification, facial expression recognition, and multi-label classification. These extensive experiments provide strong empirical evidence that adaptive gradient methods, such as Adam and RMSProp, converge to fairer minima with higher probability compared to SGD.

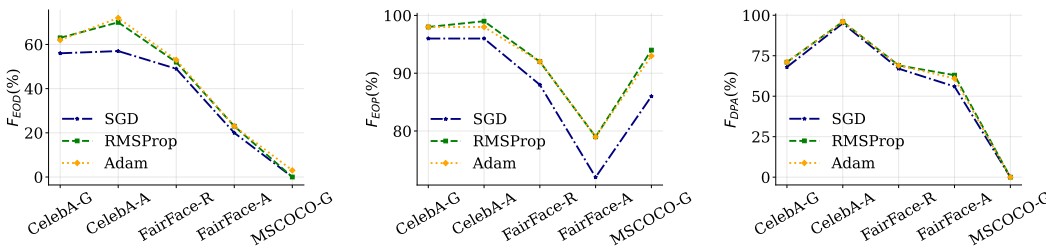

Figure 12: Fairness metrics for ResNet-50 across different datasets and sensitive attributes.

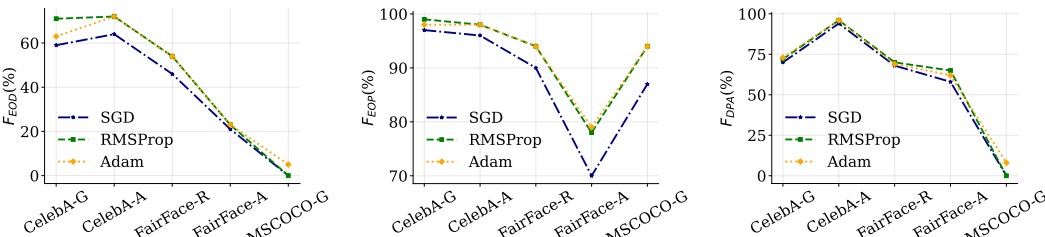

Figure 13: Fairness metrics for VGG-16 across different datasets and sensitive attributes.

## N Additional Fairness Results across Different Variants of Optimizers

Our theoretical analysis identifies gradient normalization via second moment as the key mechanism for improving group fairness. This provides a clear framework for predicting the behavior of other optimizers with regards to fairness. To validate these predictions, we have conducted a new set of experiments with SGD w/ momentum [30], AdamW [32], and AdaBound [33] variants. Following the protocol from our main experiments, we use the CelebA dataset with a ViT backbone, treating gender and age as sensitive attributes. We tuned the learning rate for each optimizer using Bayesian optimization with Hyperband and report the average fairness scores in Table 6. As shown in the table, AdamW consistently achieves the best fairness scores. This is expected, as AdamW retains the normalization mechanism of Adam and RMSProp via the second moment. In contrast, SGD w/momentum, lacking the normalization via the second moment, performs more biased. The performance of AdaBound is comparable to SGD w/momentum. The reason is that the AdaBound algorithm deliberately clips the second moments to make it act like SGD, which in turn limits the fairness advantage.

## O Additional Classification Results across Different Backbones and Datasets

Table 7 reports the accuracy and F1 scores of SGD, RMSProp, and Adam optimizers across ResNet-18, ResNet-50, and ViT backbones. We evaluate these optimizers on CelebA (facial expression recognition), MS-COCO (multi-label classification), and FairFace (gender classification) datasets. For CelebA, the performance of SGD, RMSProp, and Adam is comparable, with only minor variations. For instance, using the VGG-16 backbone, SGD achieves an accuracy of $92.22\%$, while RMSProp and Adam yield $92.37\%$ and $91.42\%$, respectively. Similarly, for MS-COCO and FairFace, the performance differences across the optimizers remain limited. The largest accuracy gap between SGD

Table 6: Fairness comparison of optimizer variants.

|  | **Gender** | | | **Age** | | |
|---|---|---|---|---|---|---|
|  | AdamW | AdaBound | SGD w/ mom. | AdamW | AdaBound | SGD w/ mom. |
| $F_{\text{EOD}}$ | 64.90 | 61.10 | 61.12 | 71.11 | 67.20 | 67.25 |
| $F_{\text{EOP}}$ | 99.90 | 96.10 | 95.07 | 98.95 | 96.60 | 96.55 |
| $F_{\text{DPA}}$ | 73.11 | 70.01 | 69.50 | 98.11 | 95.39 | 95.39 |

Table 7: Performance of different backbones across datasets. Each cell reports three numbers corresponding to SGD, RMSProp, and Adam, respectively.

| Dataset | Backbones | | | | | |
| | ResNet18 | | ResNet50 | | VGG16 | |
| | Acc. | F1 | Acc. | F1 | Acc. | F1 |
|---|---|---|---|---|---|---|
| CelebA | 91.12, 92.21, 92.19 | 91.41, 92.76, 92.80 | 91.19, 92.60, 92.39 | 91.33, 92.09, 92.25 | 92.22, 92.37, 91.42 | 92.20, 92.28, 92.50 |
| MS-COCO | 88.31, 91.10, 90.42 | 66.96, 71.26, 69.37 | 89.15, 91.89, 91.70 | 67.68, 70.48, 70.16 | 89.48, 90.11, 91.57 | 66.47, 72.04, 73.20 |
| FairFace | 86.16, 90.36, 90.31 | 87.15, 88.71, 90.30 | 88.48, 90.58, 92.41 | 89.67, 90.77, 91.57 | 91.70, 91.81, 92.09 | 90.59, 91.66, 92.69 |

and adaptive optimizers occurs on MS-COCO with the VGG-16 backbone, reaching approximately 5.5%. However, the highest disparity in fairness metrics does not occur on MS-COCO but rather on FairFace dataset. These findings indicate that the differences between SGD and adaptive optimization algorithms cannot be solely attributed to their performance. Instead, as demonstrated in Theorems 1–3, the normalization behavior of these optimizers plays a crucial role in shaping their fairness properties.

