# OpenReview forum: "Some Optimizers are More Equal: Understanding the Role of Optimizers in Group Fairness"
_NeurIPS.cc/2025/Conference — NeurIPS 2025 spotlight_

### Official Review · Reviewer_fC4c · 2025-06-01

**Clarity:** 3
**Significance:** 2
**Originality:** 2
**Rating:** 4
**Confidence:** 3

**Summary:**

This paper investigates the often-overlooked influence of optimization algorithms on group fairness in deep neural networks, particularly under severe data imbalance. Through stochastic differential equation analysis and extensive experiments on CelebA, FairFace, and MS-COCO datasets, the study reveals that adaptive optimizers like RMSProp and Adam consistently lead to fairer outcomes than stochastic methods like SGD, while maintaining comparable predictive accuracy. The research provides theoretical guarantees showing RMSProp's tendency towards fairer parameter updates and improved fairness in a single optimization step compared to SGD. Ultimately, these findings highlight the crucial role of adaptive updates as a mechanism for promoting equitable results in deep learning.

**Questions:**

1. Could the authors elaborate on how their findings complement or offer distinct advantages over existing fairness-aware approaches?
2. Could the authors either provide a stronger justification for their inclusion in the main body or consider relocating them to the Appendix for improved readability?

**Ethical Concerns:**

["NO or VERY MINOR ethics concerns only"]

**Final Justification:**

My concern has been properly addressed. However, I am still not sure whether this paper meets the standard for this venue because it does not excite and impress me particularly. Therefore, I personally maintain the score, and both acceptance and rejection of this paper is OK for me.

**Quality:**

2

**Strengths And Weaknesses:**

### Strengths

* To my knowledge, the paper presents **novel findings** concerning the inherent fairness properties of various optimizers, especially in the absence of explicit fairness constraints. This is a significant and underexplored area.
* The research effectively bolsters its core idea with a compelling blend of **intriguing theoretical analysis and extensive experimental validation**.

---

### Weaknesses

* **Proof Placement:** I recommend moving some of the theoretical proofs to the Appendix. Specifically, the proofs for Theorem 1, Theorem 2, and Theorem 3, which together occupy over a full page in the main paper, could be relocated. Given that these theorems largely rely on toy examples, such as Gaussian distributions and simplified loss functions, the proof techniques don't strike me as particularly groundbreaking. Perhaps the authors could explicitly highlight the significance of these proof skills or justify their necessity for inclusion in the main body rather than the Appendix.

* **Real-World Significance:** While the findings are undoubtedly interesting, their practical significance in real-world scenarios remains somewhat unclear. As the "Related Work" section acknowledges, numerous fairness-aware algorithms already exist, offering well-established trade-offs between fairness and accuracy. I suggest the authors elaborate further on how their findings complement or offer distinct advantages over these existing fairness-aware approaches, clarifying the unique contribution of this work to real-world applications.

* **Missing Related Work:** Several relevant papers appear to be missing from the citations. For instance, references \[1-2] also theoretically underscore the importance of debiased datasets, which aligns with some of the paper's underlying themes.

* **Extended Experimental Comparisons:** The experimental section could be expanded to include comparisons with more in-processing fairness-aware algorithms, such as those in references \[3-4]. While adding any comparative experiments would be beneficial, I am particularly keen to see whether the phenomena observed in this paper extend to these fairness-aware algorithms. For example, investigating how different optimizers influence the outcomes of a fairness-aware algorithm like the one in \[3] could significantly enhance the paper's impact within the fairness research community.

---

[1] The Implicit Fairness Criterion of Unconstrained Learning. ICML 2019.

[2] Understanding Fairness Surrogate Functions in Algorithmic Fairness. TMLR 2024.

[3] FairBatch: Batch Selection for Model Fairness. ICLR 2021.

[4] A Reductions Approach to Fair Classification. ICML 2018.

---

> ### Author Rebuttal · Authors · 2025-07-31
>
> We thank the reviewer for their valuable comments. Below we provide a careful point-by-point response to each comment. We would be happy to provide additional discussions/information in the author-reviewer discussion period.
>
> > **Proof replacement**
>
> We agree that relocating the detailed proofs can improve readability and we will revise the camera ready version of the paper accordingly. Specifically, we will move the proofs of Theorems 1, 2, and 3 to the Appendix.
>
> While we will make this change, we want to take this opportunity to clarify a key distinction and the significance of our theoretical approach. We wish to clarify that Theorems 2 and 3 are not based on the analytically tractable setup of Theorem 1. Theorems 2 and 3 establish more general results under the much broader assumption of well-behaved isotropic noisy gradient oracle with scale parameter (NGOs). Additionally, we now **relax the isotropic assumption** and establish a new theorem based on non-isotropic (anisotropic) noise. Please see our new theorem and its detailed proof in our response to the reviewer ``qYpZ''.
>
> Furthermore, the analytically tractable quadratic setup in Theorem 1 was deliberately chosen. A quadratic function serves as a powerful local approximation for any sufficiently smooth loss function near a minimum (via a second-order Taylor expansion). Therefore, our analysis in this setting offers valuable insight into the local convergence behavior of optimizers in the complex settings.
>
> > **Real-world significance and extended comparisons over fairness-aware algorithms**
>
> Our goal is not to replace these established fairness-aware methods, but rather to demonstrate that the choice of optimizer is a fundamental, orthogonal, and complementary mechanism for improving fairness. However, based on your comment, to elaborate more on the real world significance of our findings, **we have conducted a new experiment**. We test the impact of optimizers in addition to the fairness-enhancing method [1] on two new datasets (ProPublica COMPAS [2] and AdultCensus [3]) with gender as the sensitive attribute. The results are presented in the table below for SGD, Adam, and RMSProp. We repeat the process 10 times and following [3], we report the average gap in equal opportunity, equalized odds, and demographic parity below (lower is better).
>
>
> |             |        | **Equal Opportunity** |        |   |        | **Equalized Odds** |        |   |        | **Demographic Parity** |        |
> |:------------|:------:|:---------------------:|:------:|:-:|:------:|:------------------:|:------:|:-:|:------:|:----------------------:|:------:|
> | **Dataset** | Adam   | RMSProp               | SGD    | │ | Adam   | RMSProp            | SGD    | │ | Adam   | RMSProp                | SGD    |
> | &#8203;     | &nbsp; | &nbsp;                | &nbsp; | &#8203; | &nbsp; | **with fairness enhancing** | &nbsp; | &#8203; | &nbsp; | &nbsp; | &nbsp; |
> | ProPublica COMPAS  | 0.45 | 0.48 | 0.71 | │ | 2.79 | 2.78 | 2.90 | │ | 0.86 | 0.86 | 2.60 |
> | AdultCensus       | 3.15 | 3.16 | 3.38 | │ | 2.39 | 2.41 | 4.28 | │ | 7.00 | 6.80 | 11.79 |
> | &#8203;     | &nbsp; | &nbsp;                | &nbsp; | &#8203; | &nbsp; | **without fairness-enhancing** | &nbsp; | &#8203; | &nbsp; | &nbsp; | &nbsp; |
> | ProPublica COMPAS  | 13.99 | 13.90 | 15.19 | │ | 13.99 | 13.95 | 14.98 | │ | 11.49 | 11.45 | 11.80 |
> | AdultCensus       | 21.04 | 20.91 | 21.29 | │ | 20.90 | 20.91 | 21.14 | │ | 12.19 | 12.23 | 12.42
>
> As the results show, adaptive optimizers outperform SGD both with and without the explicit fairness-enhancing method. This directly leads to the contribution and practical significance of our work: choosing an adaptive optimizer is not an alternative to existing fairness algorithms, but rather a **complementary** strategy that amplifies their effectiveness. A practitioner can gain almost ``free'' and significant boost in fairness by simply selecting Adam or RMSProp as the optimizer for their chosen fairness-aware training algorithm.
>
> [1] Roh, Yuji, et al. "FairBatch: Batch Selection for Model Fairness." ICML (2021).
>
> [2] Julia Angwin, et al. "Machine bias: There’s software used
> across the country to predict future criminals. And its biased against blacks." ProPublica (2016).
>
> [3] Ron Kohavi. "Scaling up the accuracy of naive-bayes classifiers: A decision-tree hybrid." KDD (1996).
>
>
> > **Missing related work**
>
> We will add the mentioned papers to the related work section:
>
> The work by Liu et al. [1] provides a key theoretical analysis showing that standard and unconstrained machine learning implicitly favors one fairness criterion over others. The authors prove that a model's deviation from calibration is bounded by its excess risk. This is related to our work as it also investigates the fairness properties that emerge from a standard training process.
>
> The paper by Yao et al. [2] focuses on in-processing fairness methods that use surrogate functions to ensure fairness. Their theory also highlights that balanced datasets are beneficial for achieving tighter fairness and stability guarantees. This is relevant to our findings, as we also identify data imbalance as a critical factor through SDE analysis.
>
> [1] Liu et al., "The Implicit Fairness Criterion of Unconstrained Learning." ICML (2019).
>
> [2] Yao et al. "Understanding Fairness Surrogate Functions in Algorithmic Fairness." TMLR (2024).
>
>
> > **Elaboration on how findings complement or offer distinct advantages over existing fairness-aware approaches**
>
> Our core theoretical results (Theorems 2 and 3) do not make strong assumptions about the specific form of the loss function. This generality is crucial because many in-processing fairness methods operate by adding a fairness regularizer to the standard loss. Our theory suggests that even with such a modified objective, the inherent mechanics of an adaptive optimizer like Adam should still offer benefits by normalizing subgroup-specific gradient disparities more effectively than SGD.
>
> To validate this complementary effect, and motivated by your comment, we conducted **the new experiment** presented in the table in our previous response. Those results show that adaptive optimizers achieve superior fairness compared to SGD, both in a standard setup and when used as the optimization backbone for an existing fairness-enhancing algorithm. This clarifies the distinct and practical contribution of our work: the choice of an adaptive optimizer is not a substitute for dedicated fairness algorithms but is a complementary decision that can amplify their impact.
>
> > **Provide a stronger justification for inclusion of proofs in the main body or consider relocating them to the Appendix for improved readability**
>
> We agree that relocating the detailed proofs can improve the readability. For a more detailed discussion on this, please see our response to the **Proof replacement** title.

---

### Official Review · Reviewer_WGcQ · 2025-06-23

**Clarity:** 2
**Significance:** 3
**Originality:** 3
**Rating:** 4
**Confidence:** 3

**Summary:**

The paper studies the relationship between optimizers and fairness of the resulted deep learning model. By approximating SGD and RMSProp as stochastic different equations, the paper proves that RMSProp provides fairer updates than SGD and the worst-case disparities introduced in a single step by RMSProp is less than SGD. The conclusion are validated through extensive experiments across different tasks.

**Questions:**

1. Could the SDE analysis provide some insights about how RMSProp can be modified to perform even better in terms of demographic parity?

2. In what extent that Theorem 2 and Theorem 3 can be extended to the fairness metrics EOP and EOD?

3. For the proof of Lemma 1, could the relationship between Eq 43 and zero-one loss difference be elaborated more?

4. In Theorem 2, what is the definition of parameter updates between subgroups 0 and 1? Do you mean the expectation updates on the subgroups?

**Ethical Concerns:**

["NO or VERY MINOR ethics concerns only"]

**Final Justification:**

The discussion period addressed most of my concerns. The new proof the author provides relax the strong assumption on isotropic noise. And it is more clear about the relationship between gradient disparities and fairness notions now.

Even though I still feel the relationship can be studied and discussed further, the work, in my opinion is novel and valuable to be accept. So I maintain my score.

**Limitations:**

The paper does not have a limitation section. However, before the Theorems showed to be effective regarding to other fairness metrics, there should be a reminder for using the conclusion.

**Paper Formatting Concerns:**

No major formatting issues.

**Quality:**

3

**Strengths And Weaknesses:**

**Strengths**:

* It is novel to analyze how the choice of optimizers will affect demographic parity through SDE.

* The theoretical analysis is solid and the proofs are not depend on a specific network architecture.

**Weakness**:

* The paper compared the behavior between SGD and adaptive gradient methods. However, adaptive gradients methods, such as Adam, has already been widely used because of some other advantages, such as the converge speed. If the theoretical analysis cannot guide to find better optimizers regarding to fairness, the motivation seems to be weakened.

* Some of assumptions in Theorem 2 and Theorem 3 are not justified. For example, why can we assume the gradients are well-behaved isotropic NGOs.

* EOD, EOP and DPA are used in the experiments, but lack of theoretical discussion, especially when it seems like the advantage over SGD on DPA is less obvious than EOP and EOD from the experiments.

---

> ### Author Rebuttal · Authors · 2025-07-31
>
> We thank the reviewer for their valuable comments. Below we provide a careful point-by-point response to each comment. We would be happy to provide additional discussions/information in the author-reviewer discussion period.
>
> > **Adaptive methods like Adam offer advantages like speed**
>
> We agree Adam/RMSProp are popular for their speed. Our contribution is orthogonal: we explain and predict their fairness behavior. Our theory shows that per‑coordinate second‑moment scaling contracts subgroup‑specific update gaps under imbalance. This yields actionable intuition: the theory predicts that if one wants a higher fairness, they would increase the effective second‑moment strength (i.e., by tuning $\beta_2$ hyperparameter). This intuition is novel and it is not related to the well-known advantages of the adaptive methods.
>
>
> > **Assumption of isotropic noise in Theorems 2 and 3**
>
> Please kindly note that the assumption of isotropic noise is not uncommon in the literature. For instance, recent work such as [1] uses an isotropic noise assumption to illustrate why decaying learning rate during training is an effective mechanism for getting to a better minima.
>
> However, inspired by the reviewer's comment, **we now relax the conditions in Theorems 2 and 3 to non-isotropic (anisotropic) noise**. We show that the results are correct even in the anisotropic noise case.
>
> **Theorem 2 for anisotropic noise**: Consider a population that consists of two subgroups with subgroup-specific loss functions $\mathcal{L}_0(w)$ and $\mathcal{L}_1(w)$, sampled with probabilities $p_0$ and $p_1$, respectively. Suppose a stochastic (online) training regime, in which each parameter update is computed from a sample drawn from one of the two subgroups. Suppose the gradients $\nabla \mathcal{L}_0(w_k)$ and $\nabla \mathcal{L}_1(w_k)$ are well-behaved anisotropic NGOs, namely $\mathcal{N}(\mu_0, \Sigma_0)$ and $\mathcal{N}(\mu_1, \Sigma_1)$, respectively.
> Then, the difference in parameter updates between subgroups 0 and 1 under RMSProp has an upper bound given by the corresponding difference under SGD.
> Consequently, RMSProp offers fairer updates across subgroups.
>
> *Proof*: For SGD, the difference between the parameter updates across subgroups is $||\nabla \mathcal{L}_0 - \nabla \mathcal{L}_1||$. Regarding RMSProp, let's compute the expected value and the variance of $v_k$:
>
> $$
>     \mathbb{E}[v_k] = \gamma^k v_0 + (1-\gamma^k) [ p_0 \left( \mu_0 ^2 + diag(\Sigma_0) \right)+ p_1 \left( \mu_1^2 + diag(\Sigma_1) \right)].
> $$
>
> Where $diag(\Sigma_0)$ and $diag(\Sigma_1)$ are the vectors with the diagonal elements of $\Sigma_0$ and $\Sigma_1$, respectively. Similar to the the isotropic case, we can show that the variance of $v_k$ is negligible and thus $v_k \rightarrow E[v_k]$. Hence, the difference between parameter updates are $|| D (\nabla \mathcal{L}_0 - \nabla \mathcal{L}_1) ||$, where $D$ is a diagonal matrix, whose $jj$th element can be computed as:
>
> $$
>      D_{jj} =  \frac{1}{\sqrt{p_0 \left( \mu_{0_j} ^2 + \sigma_{0_j}^2 \right)+ p_1 \left( \mu_{1_j}^2 + \sigma_{1_j}^2  \right) + \epsilon}}.
> $$
>
> Prior work [2] has experimentally shown that $\sigma^2 > \mu^2$. Additionally, in stochastic (online) training setup: $\Theta^2 \rightarrow1$. Thus, $D_{jj}<1$. This completes the proof.
>
>
> The extension of Theorem 3 follows directly as the proof of Theorem 3 relies on the conclusion of Theorem 2.
>
> We will add this to the appendix of the final version of our paper.
>
> [1] Shi, Bin, et al. "On learning rates and Schrödinger operators." Journal of Machine Learning Research (2023).
>
> [2] Malladi, Sadhika, et al. "On the SDEs and scaling rules for adaptive gradient algorithms." Advances in Neural Information Processing Systems (2022).
>
>
> > **Advantage over SGD on DPA is less obvious than EOP and EOD from the experiments.**
>
> Fig. 2 shows that the advantage of adaptive optimizers over SGD on DPA is less obvious. This empirical result actually provides the confirmation of our Theorem 1. This theorem predicts that the DPA advantage of RMSProp is most significant under severe data imbalance. In datasets with only moderate imbalance (e.g., CelebA-Gender, with a 42\% male-to-all ratio), the DPA gap is smaller. However, the DPA advantage is more obvious in the case of FairFace with race sensitive attribute, as it exhibits extreme imbalance, with a minority-to-all ratio of approximately 0.9\%.
>
> > **How can RMSProp be modified to perform even better in terms of demographic parity?**
>
> Our SDE analysis provides a path for some modifications in RMSProp to further enhance its fairness. Our analysis (culminating in Eq. 15 and 20) pinpoints that the final model's bias stems from the SDE's drift term, which is driven by the imbalanced sampling probabilities ($p_0$ and $p_1$) of the data. Additionally, RMSProp's relative advantage comes from how it scales the updates: The denominator of the update involves the term $u_t$, which helps to avoid the domination by the majority sub-group. Based on these insights, a direct path to improving RMSProp's fairness is to correct the biased drift term while retaining the benefits of its adaptive normalization. The SDE model suggests that if we could modify the algorithm to optimize a debiased loss function, the stationary distribution of our SDE would shift its center to the truly fair minimum. This modification would use a debiased gradient in each step, computed by re-weighting subgroups within a mini-batch to simulate a balanced population. Our SDE framework predicts this change would directly correct the drift term, shifting the stationary distribution to be centered at the fair optimum.
>
> > **In what extent that Theorem 2 and Theorem 3 can be extended to the fairness metrics EOP and EOD?**
>
> Please kindly note that Theorem 2 is *agnostic to the fairness metrics*. Theorem 2 is broadly applicable and provides the core intuition for the improvements in EOD and EOP. Its proof focuses on the fundamental mechanism of the optimizer that RMSProp shrinks the disparities between subgroup gradients ($||\nabla\mathcal{L}_0 - \nabla\mathcal{L}_1||$). By preventing the model from being disproportionately pulled toward one subgroup's minimum, it promotes more balanced error rates (e.g., true positive and false positive rates) across all groups. Since EOD and EOP are measures of error rate equality, the general mechanism established in Theorem 2 directly explains the strong empirical improvements we observe for these metrics.
>
> Theorem 3 relies on the DPA gap.  Direct formal extension to EOD or EOP is highly challenging, as it requires analyzing a fairness gap conditioned on the true label. This conditioning complicates the gradient analysis significantly and is beyond the current scope of the paper.
>
>
> > **For the proof of Lemma 1, could the relationship between Eq 43 and zero-one loss difference be elaborated more?**
>
> Eq. 43 shows the definition of the demographic parity. Directly optimizing the non-differentiable demographic parity gap is intractable, thus it is a standard practice to replace it with a tractable setup [1]. Following prior works, we replace this with the gap in the loss functions for the sub-groups. The principle is that if a model yields a similar loss distribution for two sub groups, it is being penalized similarly for its mistakes on both, which encourages its output behavior to align.
>
> [1] Li et al. "Achieving fairness at no utility cost via data reweighing with influence." ICML (2022).
>
> > **The definition of parameter updates in Theorem. 2**
>
> In Theorem 2, the "difference in parameter updates" refers to a difference in two updates: one that is occurred by drawing the samples from sub group 0 and the one by drawing samples from sub group 1. We analyze the difference that would occur if the sample drawn at iteration $k$ came from subgroup 0 versus if it came from subgroup 1. As an example, For SGD, the update rule is $w_{k+1} = w_{k} - \eta \nabla\mathcal{L}$; thus if a sample is drawn from subgroup say 0, the update relative to $w_k$ is $\nabla\mathcal{L}_0$. If there is imbalance in the dataset (e.g., the chance of drawing one sub group is substantially higher), the gradients (updates) from the majority would dominate the optimization process.

---

> > ### Comment · Reviewer_WGcQ · 2025-08-01
> >
> > Thank you for the authors' valuable response. The author addressed most of my concerns. I have a little problem with Theorem 2 which seems to prove that the gradients disparities between any two groups will be reduced. However, in some times, different fairness metrics such as demographic parity and equal opportunity could be contradict to each other.
> >
> > Does this mean the gradients disparities may failed to reflect some fairness metrics or is there some implicit assumption to select two groups?

---

> > > ### Author Response · Authors · 2025-08-02
> > >
> > > We sincerely thank the reviewer for promptly reading our rebuttal and for their thoughtful question. The reviewer correctly highlights the known challenge that different fairness metrics can sometimes be in tension. Nowhere in the literature has a universal fairness guarantee been established [1, 2].
> > >
> > > The reviewer asks how reducing gradient disparities (Theorem 2) relates to fairness, especially when metrics might conflict. Our response is twofold:
> > >
> > > **1.** Theorem 2 states that adaptive methods mitigate the risk of the training process being dominated by gradients from the majority subgroup and ending up in a minima closer to the optimum minima of that subgroup. This is crucial for achieving fairness under any **performance-based** fairness metric (like disparities in accuracy among sub-groups). If the updates are consistently dominated by the majority subgroup, the model will inevitably converge to a minimum that is optimal for that subgroup, leading to larger performance gaps and thus worse fairness.
> > >
> > > **2.** We use the results of Theorem 2 to provide a formal link to demographic parity as a specific fairness criterion. Theorem 3 explicitly uses the gradient-shrinking property established in Theorem 2 to prove that RMSProp suffers a smaller worst-case increase in the demographic parity gap per step than SGD. This demonstrates how the dynamic of "fairer update" translates into a tangible benefit for this fairness definition.
> > >
> > > As we state in the paper (l. 267-269), we acknowledge that this single mechanism does not resolve all tensions between all metrics. However, our extensive empirical results (Fig. 2, Fig. 11-13 in the paper) and our new experiments using tabular data (please see our response to Reviewer "fC4c" under ``real-world significance’’) provide more evidence that, in practice, adaptive optimizers help improve fairness across three different metrics.
> > >
> > >
> > > [1] Kleinberg, Jon, Sendhil Mullainathan, and Manish Raghavan. "Inherent trade-offs in the fair determination of risk scores." arXiv preprint arXiv:1609.05807 (2016).
> > >
> > > [2] Pleiss, Geoff, et al. "On fairness and calibration." NeurIPS (2017).

---

> > > > ### Comment · Reviewer_WGcQ · 2025-08-04
> > > >
> > > > Thank you again for your thoughtful reply. I will maintain my score to accept.

---

### Official Review · Reviewer_qYpZ · 2025-06-28

**Clarity:** 3
**Significance:** 3
**Originality:** 3
**Rating:** 4
**Confidence:** 3

**Summary:**

This paper tackles a crucial yet under-explored question in fair machine learning: How does the choice of optimization algorithm affect group fairness in deep neural networks?  It develops an SDE-based theoretical framework, proves two theorems bounding fairness gaps, and empirically evaluates on three vision datasets across multiple fairness metrics. The main claim is that adaptive optimizers consistently find fairer minima without sacrificing accuracy.

The main contributions of this paper include:

1) It proposed a solid SDE-based approach, which uses SDEs and Fokker-Planck equation to connect optimizer dynamics to fairness outcomes. It includes two concrete theorems and closed-form distributions.
2) Practical impact. No need to change model architecture or loss, simply choosing RMSProp/Adam over SGD can improve fairness, which lowers the barrier for real-world adoption .
3) Open-source release for reproducibility. The authors publicly released the code and training pipelines at an anonymized repository, so that the reported experimental results can be reproduced.

**Questions:**

How well do isotropic NGOS assumptions match actual gradient noise in large models?

As to "We hope our work will encourage broader adoption of adaptive optimizers in fairness-critical domains" , since all experiments are conducted using vision datasets, what is the chance of the conclusions can be applied to other modalities?

Does the fairness conclusion depend on the choices of learning rates, decay rates, and batch sizes? If yes, how to choose proper hyperparemeters for this study?

**Ethical Concerns:**

["NO or VERY MINOR ethics concerns only"]

**Final Justification:**

The detailed rebuttal is appreciated. The additional analysis of various SGD variants provides meaningful insights, and the extension of Theorems 2 and 3 is a noteworthy contribution. The revised version represents a substantial improvement over the original submission, and so I have updated my review accordingly.

**Limitations:**

Yes.

**Paper Formatting Concerns:**

No.

**Quality:**

3

**Strengths And Weaknesses:**

The main strengths are
1) Novel theoretical analysis. The paper provides the first SDE-based analysis that connects optimizer noise dynamics directly to fairness.
2) Provable fairness bounds: The two theorems offer sound and general guarantees about optimizer-induced fairness effects, which is beyond empirical observations .
3) Solid experimental results. Experiments use three datasets, three tasks, multiple architectures, three fairness metrics, and include statistical testing and imbalance-controlled ablations, solidifying the practical relevance.

The main weaknesses are
1) The core SDE analysis is based on a one-dimensional quadratic loss with two classes. Deep networks are often high-dimensional and for multi-class, so it is unclear how these results extend.
2) Comparisons are only between SGD and RMSProp/Adam. There are other popular variants such as SGD with momentum, AdaBound, AdamW, etc. that are not considered.
3) Theorems 2 and 3 assume isotropic, well-behaved NGOS noise, which is an ideal condition that lacks empirical justification.

---

> ### Author Rebuttal · Authors · 2025-07-31
>
> We thank the reviewer for their valuable comments. Below we provide a careful point-by-point response to each comment. We would be happy to provide additional discussions/information in the author-reviewer discussion period.
>
> > **SDE analysis (Theorem 1) based on a one-dimensional quadratic loss**
>
> We deliberately designed a one-dimensional quadratic case and studied it as a warm‑up setup for 2 main purposes: *1.*  The local dynamics around any sufficiently smooth minimum can be approximated by a quadratic function via Taylor expansion. Hence, our SDE analysis of the fair minima provides crucial insights into the local convergence behavior of the optimizers on more general cases. *2.* To provide a clear and unambiguous *intuition* about the fairness behavior of adaptive and SGD optimizers that would be obscured in complex settings.
>
> Also, please kindly note that we build upon the intuition from this tractable setup with more general, high-dimensional theorems. Our main theoretical results in Section 3.3 (Theorems 2 and 3) **are not limited to the 1D quadratic case.**
>
> > **Other variants such as SGD with momentum, AdamW, and AdaBound**
>
> Our theoretical analysis (Theorems 1-3) identifies gradient normalization via second moment as the key mechanism for improving group fairness. This provides a clear framework for predicting the behavior of other optimizers. To validate these predictions, we have conducted a new set of experiments with SGD w/momentum, AdamW, and AdaBound. Following the protocol from our main experiments, we use the CelebA dataset with a ViT backbone, treating gender and age as sensitive attributes. We tuned the learning rate for each optimizer using Bayesian optimization with Hyperband and report the average fairness scores below (higher is better).
>
> |               |        | **Gender** |        |   |        | **Age** |        |
> |:--------------|:------:|:----------:|:------:|:-:|:------:|:-------:|:------:|
> |  | AdamW  | AdaBound   | SGD w/mom.    | │ | AdamW  | AdaBound| SGD w/mom.    |
> | $F_{EOD}$         |   64.90    | 61.10          | 61.12      | │ | 71.11      |    67.20    | 67.25      |
> | $F_{EOP}$         | 99.90      | 96.10          | 95.07      | │ | 98.95      | 96.60       | 96.55      |
> | $F_{DPA}$         | 73.11      | 70.01          | 69.50      | │ | 98.11      | 95.39       | 95.39      |
>
>
> As shown in the table, AdamW consistently achieves the best fairness scores. This is expected, as AdamW retains the normalization mechanism of Adam and RMSProp via the second moment. In contrast, SGD w/momentum, lacking the normalization via the second moment, performs more biased. The performance of AdaBound is comparable to SGD w/momentum. The reason is that the AdaBound algorithm deliberately clips the second moments to make it act like SGD, which in turn limits the fairness advantage.
>
>
> > **Assumption of isotropic noisy gradient oracle with scale parameter (NGOS) in Theorems 2 and 3**
>
> Please kindly note that the isotropic assumption is common in the literature for analyzing optimizer dynamics as it yields clean and interpretable results. For instance, recent work such as [1] also employs an isotropic noise assumption to illustrate why decaying learning rate during the training is an effective mechanism.
>
> However, inspired by the reviewer's comment, **we now relax the conditions in Theorems 2 and 3 to non-isotropic (anisotropic) noise.** We show that the results are correct even in the anisotropic case.
>
> **Theorem 2 for anisotropic noise**: Consider a population that consists of two subgroups with subgroup-specific loss functions $\mathcal{L}_0(w)$ and $\mathcal{L}_1(w)$, sampled with probabilities $p_0$ and $p_1$, respectively. Suppose a stochastic (online) training regime, in which each parameter update is computed from a sample drawn from one of the two subgroups. Suppose the gradients $\nabla \mathcal{L}_0(w_k)$ and $\nabla \mathcal{L}_1(w_k)$ are well-behaved anisotropic NGOs, namely $\mathcal{N}(\mu_0, \Sigma_0)$ and $\mathcal{N}(\mu_1, \Sigma_1)$, respectively.
> Then, the difference in parameter updates between subgroups 0 and 1 under RMSProp has an upper bound given by the corresponding difference under SGD.
> Consequently, RMSProp offers fairer updates across subgroups.
>
> *Proof*: For SGD, the difference between the parameter updates across subgroups is $||\nabla \mathcal{L}_0 - \nabla \mathcal{L}_1||$. Regarding RMSProp, let's compute the expected value and the variance of $v_k$:
>
> $$
> \mathbb{E}[v_k] = \gamma^k v_0 + (1-\gamma^k) [ p_0 \left( \mu_0 ^2 + diag(\Sigma_0) \right)+ p_1 \left( \mu_1^2 + diag(\Sigma_1) \right)].
> $$
>
> Where $diag(\Sigma_0)$ and $diag(\Sigma_1)$ are the vectors with the diagonal elements of $\Sigma_0$ and $\Sigma_1$, respectively. Similar to the the isotropic case, we can show that the variance of $v_k$ is negligible and thus $v_k \rightarrow E[v_k]$. Hence, the difference between parameter updates are $|| D (\nabla \mathcal{L}_0 - \nabla \mathcal{L}_1) ||$, where $D$ is a diagonal matrix, whose $jj$th element can be computed as:
>
> $$
> D_{jj} =  \frac{1}{\sqrt{p_0 \left( \mu_{0_j} ^2 + \sigma_{0_j}^2 \right)+ p_1 \left( \mu_{1_j}^2 + \sigma_{1_j}^2  \right) + \epsilon}}.
> $$
>
> Prior work [2] has experimentally shown that $\sigma^2 > \mu^2$. Additionally, in stochastic (online) training setup: $\Theta^2 \rightarrow1$. Thus, $D_{jj}<1$. This completes the proof.
>
>
> The extension of Theorem 3 follows directly as the proof of Theorem 3 relies on the conclusion of Theorem 2.
>
> We will add this to the appendix of the final version of our paper.
>
> [1] Shi, Bin, et al. "On learning rates and Schrödinger operators." Journal of Machine Learning Research (2023).
>
> [2] Malladi, Sadhika, et al. "On the SDEs and scaling rules for adaptive gradient algorithms." Advances in Neural Information Processing Systems (2022).
>
> > **What is the chance that the conclusions can be applied to other modalities beyond vision?**
>
> Please kindly note that our claim is not modality-specific. Theorems 1, 2, and 3 reason at the level of optimizer dynamics under subgroup imbalance. They conclude that adaptive optimizer's help enhance the group fairness regardless of the input modality. These arguments depend on the geometry of gradient updates, not on the input.
>
> However, as per the reviewer's comment, we now perform a quick experiment using a new modality: **tabular data**. We use ProPublica COMPAS [1] and AdultCensus [2] datasets with gender as the sensitive attribute. We do classification using logistic regression, while optimizing the learning rate using Bayesian optimization. We repeat the process 10 times and following [3], we report the average gap in equal opportunity, equalized odds, and demographic parity below (lower is better).
>
> |             |        | **Equal Opportunity** |        |   |        | **Equalized Odds** |        |   |        | **Demographic Parity** |        |
> |:------------|:------:|:---------------------:|:------:|:-:|:------:|:------------------:|:------:|:-:|:------:|:----------------------:|:------:|
> | **Dataset** | Adam   | RMSProp               | SGD    | │ | Adam   | RMSProp            | SGD    | │ | Adam   | RMSProp                | SGD    |
> | ProPublica COMPAS  | 13.99 | 13.90 | 15.19 | │ | 13.99 | 13.95 | 14.98 | │ | 11.49 | 11.45 | 11.80 |
> | AdultCensus       | 21.04 | 20.91 | 21.29 | │ | 20.90 | 20.91 | 21.14 | │ | 12.19 | 12.23 | 12.42 |
>
>
> As shown, adaptive optimizers offer better fairness over SGD in all cases. This confirms that our results are not limited to the vision application.
>
> [1] Julia Angwin, et al. "Machine bias: There’s software used
> across the country to predict future criminals. And its biased against blacks." ProPublica (2016).
>
> [2] Ron Kohavi. "Scaling up the accuracy of naive-bayes classifiers: A decision-tree hybrid." KDD (1996).
>
> [3] Roh, Yuji, et al. "FairBatch: Batch Selection for Model Fairness." ICML (2021).
>
>
> > **Fairness conclusion depends on the choices of learning rates and other hyperparameters**
>
> Indeed the fairness outcomes can be influenced by hyperparameter choices, which is why we adopted a highly rigorous approach to mitigate this potential influence. Our primary method for ensuring a fair comparison was a robust and automated search for the most critical hyperparameter: the learning rate. As detailed in our Appendix J (l. 679) and F (l. 600), we did not manually select these values. Instead, we employed Bayesian Optimization with Hyperband algorithm to systematically find the best learning rate for each optimizer independently. We then run each optimizer 2 times with their tuned learning rate and report the average results. For other hyperparameters, we use the standard PyTorch default values. However, to further address the reviewer's comment, we conduct an additional hyperparameter search on the CelebA dataset with gender and age as the sensitive attributes using ViT backbone. This new set of experiments goes beyond only tuning the learning rate and uses Bayesian Optimization with Hyperband to jointly optimize learning rate, decay rate, and batch size.
>
> |               |        | **Gender** |        |   |        | **Age** |        |
> |:--------------|:------:|:----------:|:------:|:-:|:------:|:-------:|:------:|
> |  | Adam  | RMSProp   | SGD    | │ | Adam  | RMSProp| SGD    |
> | $F_{EOD}$         |   65.21   | 65.18          | 62.66      | │ | 72.34      |    71.99    | 68.40      |
> | $F_{EOP}$         | 99.90      | 99.91          | 96.60      | │ | 99.10      | 99.10       | 97.77      |
> | $F_{DPA}$         | 73.50      | 73.68          | 60.80      | │ | 98.20      | 98.20       | 95.40      |
>
>
> The pattern in the above table is consistent with our initial findings. In all the scenarios, adaptive optimizers offer better fairness.

---

> > ### Comment · Reviewer_qYpZ · 2025-08-05
> >
> > The detailed rebuttal is appreciated. The additional analysis of various SGD variants provides meaningful insights, and the extension of Theorems 2 and 3 is a noteworthy contribution. The revised manuscript represents a substantial improvement over the original submission, and so I have updated my review accordingly.

---

> > > ### Author Response · Authors · 2025-08-05
> > >
> > > We sincerely thank the reviewer for the thoughtful re-evaluation of our work and for updating their score. We are especially grateful for their constructive feedback.  We remain fully available and would be happy to elaborate further on any aspect of our work.

---

### Official Review · Reviewer_htpx · 2025-07-03

**Clarity:** 3
**Significance:** 3
**Originality:** 4
**Rating:** 5
**Confidence:** 3

**Summary:**

In machine learning, optimization algorithms adjust the parameters to minimize the loss function. What hasn’t been examined yet is what effect optimizers have on common group fairness criteria. This paper sets out to do so by first examining the role of optimizers in an analytically tractable setup. The authors focus on two categories of optimization algorithms, namely adaptive ones and stochastic ones. In their theoretical examination, they focus on RMSProp (an adaptive optimization algorithm) and SGD (which is stochastic). They use stochastic differential equation analysis (SDE) to find that RMSProp leads to fairer updates than SGD (specifically, the difference in parameter updates between the demographic groups is smaller). The authors then confirm this result experimentally using three datasets and three group fairness definitions (demographic parity, equality of opportunity and equalized odds). The better performance of adaptive methods on group fairness holds not only when the adaptive methods lead to better accuracy. Adaptive optimizers can thus be a simple and cost-effective way to improve group fairness criteria (before potentially using other methods to improve fairness further).

**Questions:**

- Why do you use demographic parity as a fairness metric for a gender classification system? In gender classification, we care about correct labels, not whether the likelihood of getting a certain label is the same for all demographic groups. If one group is underrepresented in the dataset, there’s no reason (at least from my point of view) to expect demographic parity (that’s different for cases where, in an ideal world, we would expect different groups to have the same distribution for the ground truth label).
- Since simply changing the optimizer method does not give us any fairness guarantees, do you see this change in optimizer to be used with other methods (e.g., post-processing methods)?
- Out of curiosity (so not saying it’s missing): Have you considered examining other group fairness criteria, such as sufficiency (predictive parity)? If so, why did you decide against it?
- Fig. 1: Why is SGD performing so much better early on at about 20 epochs?

**Ethical Concerns:**

["NO or VERY MINOR ethics concerns only"]

**Final Justification:**

The rebuttal answered my open questions and the paper makes an interesting and novel contribution, so I'll maintain my positive score.

**Limitations:**

Yes

**Paper Formatting Concerns:**

/

**Quality:**

4

**Strengths And Weaknesses:**

Strengths labeled with [S], weaknesses with [W], general thoughts/comments with [T]

# Major
## Introduction
- [S] Well-written introduction that gets the point across
## Related work
- [S] My expertise is in fairness and not in theoretical ML/optimizers, but I thought the fairness section of related work was well-written and the section on optimizers helped me remember some basic stuff
- [T] l.74f.: The reason why you focus on group fairness is plausible, although I wonder to what extend individual fairness would even be implementable for the facial recognition dataset
## Method
- [S] I appreciate the section on preliminaries to create a shared understanding
- [W] However, building some more intuition would have really helped me get the paper content faster. The explanations are still quite theoretical and based on formulas, which didn’t help me in getting a good intuition for what’s happening. For something like the NGOs, for example, explaining in basic terms what it does would have helped me a lot rather than jumping straight to the formula.
- [S] I appreciate that theoretical results were followed by empirical tests as sanity checks and a more robust verification

## Experiments
- [W] What does fairness in the context of gender classification mean? Is a fair classification simply a correct one? If so, why are you using demographic parity, equalized odds and equal opportunity over just accuracy? In other words: The experimental results from these datasets are interesting if you see the dataset as just a stand-in for a dataset with sensitive attributes, but using these fairness metrics for these datasets implicitly implies that it makes sense to measure fairness this way for real-world gender prediction – which I think is not the case, so this should be contextualized.

# Minor
## Related work
- l.83: typo in “learning algorithm”
- l.88f.: “One major limitation is their computational overhead or interference 89 with the training phase.” → Post-processing, when implemented as a simple threshold rule, for example, can be fairly cheap and easy to implement. However, your point still stands about the ease of changing the optimizer.
- l.123: add “(ODE)” abbreviation after “ordinary differential equations” as you use the abbreviation in line 126 without introducing it
## Experiments
- Fig. 2: Please don’t use a line chart here as it implies a continuous or ordered relationship between data points when the CelebA-G, CelebA-A, etc. datapoints don’t have a continuous relationship.
- l.307: The fairness metrics (F_{EOD} and the others) are only defined in the appendix. I don’t think that’s ideal since it’s an important part of the evaluation, but I can live with it. However, it seems important to at least clarify the naming convention (F_{EOD} = equalized odds, F_{EOP} = equality of opportunity and F_{DPA} = demographic parity) in the main text once.
- As you don’t explain how F_{EOD} and the other fairness criteria are calculated in the main text, it would be helpful to at least mention whether you want these values to be high or low. This helps the reader interpret Fig. 2 more quickly.
## Conclusion
- l. 389: typo in “demographic disparity”

---

> ### Author Rebuttal · Authors · 2025-07-31
>
> We thank the reviewer for their valuable comments. Below we provide a careful point-by-point response to each comment. We would be happy to provide additional discussions/information in the author-reviewer discussion period.
>
> > **Individual fairness feasibility in facial recognition**
>
> Individual fairness implies that the model should treat similar individuals, similarly. Given that each individual is different in FR datasets, this would not be very meaningful. However, one could define some metric or parameter, according to which different individuals are similar. This metric could then be utilized to ensure fairness for two individuals who are close in terms of that metric. Having said this, this approach is not quite common in FR tasks.
>
> > **Intuition about noisy gradient oracle with scale parameter (NGOS)**
>
> NGOS is simply a formal way to model the gradients used in mini-batch training. Because gradients are computed on a small batch of data rather than the entire dataset, the gradient at each step is not the ``true'' gradient, rather it is the true gradient plus some random noise from the sampling process. NGOS captures this by representing the stochastic gradient as the sum of the true gradient and a noise term. We will add this description to the text of the paper.
>
> > **Why demographic parity for gender classification?**
>
> Please kindly note that in the gender classification experiment, while the prediction target (output class) is gender, the *sensitive attributes* we evaluate for fairness are race and age, not gender itself (please see l. 665). Thus, demographic parity in our work measures whether the model predicts a specific gender with the same probability across different racial or age groups. As an example, with regards to demographic parity, we are interested in the fact that the probability of being "female" is the same across "young" and "old" subgroups.
>
> > **Typos**
>
> We thank the reviewer for catching these typos. We will correct all the noted typos and perform a thorough proofreading to prevent recurrences in the final version.
>
> > **Clarifying ``computational overhead'' of fairness methods vs. optimizer switch**
>
> We agree that some fairness interventions, such as simple post-processing rules, can be computationally inexpensive. However, as the reviewer correctly noted, our intention with the statement on l.88 was to highlight a contrast. Specifically, we aimed to compare the introduction of a separate fairness-enhancing module with the inherent simplicity of selecting a different optimizer. This distinction is important for practitioners who are looking for a straightforward way to improve fairness in their pipeline. We will revise l.88 to reflect this.
>
> > **Usage of line charts to represent the results**
>
> We will replace the line chart in Figure 2 with a grouped bar chart to more accurately represent the data. We will also apply this correction to Figures 11, 12, and 13 in the appendix to ensure consistency.
>
> > **Placement and clarification of fairness metrics**
>
> As the reviewer correctly points out, the definitions of fairness metrics are currently in Appendix G, and the clarification that higher values are better is in l.629. We agree that moving them to the main text would help the readers to more easily understand the figures. We will do so in the final version, using the extra page allowed for the final version.
>
> > **Change in optimizer to be used with other fairness-enhancing methods**
>
> Our Theorems 2 and 3 are general with respect to the loss function, which suggests their fairness benefits should be complementary to existing fairness-enhancing methods that add regularization terms to the primary loss. To validate this hypothesis, and inspired by feedback from this reviewer and reviewer ``fC4c'', we have now conducted a new experiment where we test the impact of optimizers in addition to the fairness-enhancing method [1] on two new datasets (ProPublica COMPAS [2] and AdultCensus [3]) with gender as the sensitive attribute. The results are presented in the table below for SGD, Adam, and RMSProp. Following [1], we report the average gap in equal opportunity, equalized odds, and demographic parity below (lower is better)vover 10 runs.
>
>
> |             |        | **Equal Opportunity** |        |   |        | **Equalized Odds** |        |   |        | **Demographic Parity** |        |
> |:------------|:------:|:---------------------:|:------:|:-:|:------:|:------------------:|:------:|:-:|:------:|:----------------------:|:------:|
> | **Dataset** | Adam   | RMSProp               | SGD    | │ | Adam   | RMSProp            | SGD    | │ | Adam   | RMSProp                | SGD    |
> | &#8203;     | &nbsp; | &nbsp;                | &nbsp; | &#8203; | &nbsp; | **with fairness enhancing** | &nbsp; | &#8203; | &nbsp; | &nbsp; | &nbsp; |
> | ProPublica COMPAS  | 0.45 | 0.48 | 0.71 | │ | 2.79 | 2.78 | 2.90 | │ | 0.86 | 0.86 | 2.60 |
> | AdultCensus       | 3.15 | 3.16 | 3.38 | │ | 2.39 | 2.41 | 4.28 | │ | 7.00 | 6.80 | 11.79 |
> | &#8203;     | &nbsp; | &nbsp;                | &nbsp; | &#8203; | &nbsp; | **without fairness-enhancing** | &nbsp; | &#8203; | &nbsp; | &nbsp; | &nbsp; |
> | ProPublica COMPAS  | 13.99 | 13.90 | 15.19 | │ | 13.99 | 13.95 | 14.98 | │ | 11.49 | 11.45 | 11.80 |
> | AdultCensus       | 21.04 | 20.91 | 21.29 | │ | 20.90 | 20.91 | 21.14 | │ | 12.19 | 12.23 | 12.42 |
>
> Regarding the fairness enhancing method, across all metrics on both datasets, Adam and RMSProp achieve substantially lower (better) fairness gaps than SGD. This experiment demonstrates that the fairness benefits of adaptive optimizers are not limited to standard training but are complementary to and can amplify the effectiveness of existing fairness interventions. This finding has considerable impact, and we will add this new experiment and analysis to the final version of the paper. We also report the case of not using the fairness enhancing method, in which, as expected, Adam and RMSProp achieve better fairness.
>
> [1] Roh, Yuji, et al. "FairBatch: Batch Selection for Model Fairness." ICML (2021).
>
> [2] Julia Angwin, et al. "Machine bias: There’s software used
> across the country to predict future criminals. And its biased against blacks." ProPublica (2016).
>
> [3] Ron Kohavi. "Scaling up the accuracy of naive-bayes classifiers: A decision-tree hybrid." KDD (1996).
>
>
>
> > **Have you considered examining other group fairness criteria, such as sufficiency (predictive parity)?**
>
> Across recent work, the most frequently used group fairness criteria are equalized odds, equal opportunity, and demographic parity, whereas sufficiency (predictive parity/calibration) is reported less often [1]. We therefore adopted these prevalent criteria as our primary objectives.
>
> [1] Liu, Mingxuan, et al. "A scoping review and evidence gap analysis of clinical AI fairness." npj Digital Medicine 8.1 (2025): 360.
>
> > **In Fig. 1, why is SGD performing so much better early on at about 20 epochs?**
>
> Fig. 1 shows the fraction of 1000 independent runs whose iterate lies inside a fair neighborhood (near the fair minima of $w$=0) in the presence of both mild and severe imbalance. The apparent early lead of SGD at first 20 epochs is a transient crossing effect, not stable convergence. Because SGD has larger effective drift and diffusion terms (please see the stochastic differential equation of SGD in Eq. 11) many trajectories pass through the fair neighborhood early on as they move toward the minima that happens to be less fair. This inflates the early within-fair-neighborhood counts, but those trajectories do not remain there. In contrast, RMSProp’s diagonal normalization slows entry but retains more mass near the fair neighborhood. We will add this discussion to the final version of the paper.

---

> > ### Comment · Reviewer_htpx · 2025-08-03
> >
> > Thank you for taking the time to reply to my concerns so carefully - I appreciate it!
> >
> > Let me reply point-by-point to the points I want to reply to. Thanks for your thoughts/clarification on the other points as well!
> >
> > ---
> >
> > *Individual fairness feasibility in facial recognition*
> >
> > Right, defining a similarity metric would be possible, but I’m unsure whether similarity between faces could really be defined without echoing pseudoscientific and highly problematic ideas like physiognomy. But luckily, that’s not what your paper is about :)
> >
> > *Why demographic parity for gender classification?*
> >
> > Thanks, I got that, but I think my concern was actually a different one (if I’m not misinterpreting your reply). I think I didn’t express my concern well (as I also had to take a moment to reflect on it again when I read my own comment – apologies for that!), so I’ll try to do a better job this time. My understanding is that you evaluate demographic parity on a data sample that you feed to the classifier (e.g., your test data). So you pool the outputs from the model for different groups (let’s say different racial groups). Then you compare the probability of being classified as “female” for a person in the group of Black people to the probability of being classified as “female” for a person in the group of white people – correct? But what if there are considerably more women in one group than the other? Depending on how your dataset is collected, this could happen. (Technically, the same argument applies for every classifier, but usually, we can safely assume that the training data is representative. For facial recognition tasks, I’m not sure how representative the datasets are – especially considering that the reason why gender classifiers often work less for minorities is that the training data is not representative.) So why do you use demographic parity? Do you just assume that the distribution of gender labels is equal between, say, racial groups in the dataset?
> >
> > *Change in optimizer to be used with other fairness-enhancing methods*
> >
> > Thanks for sharing the additional experimental results, that’s very interesting! There’s just one point I made previously that I’d like to reiterate: The experimental results from these datasets are interesting if you see the dataset as just a stand-in for a dataset with sensitive attributes, but using these fairness metrics for these datasets implicitly implies that it makes sense to measure fairness this way for real-world [classifiers in this context] – which I think is not the case, so this should be contextualized.
> > I’m adding this because for COMPAS, for example, we probably don’t want equal incarceration rates of men and women, given that men commit more crimes. I think it would be helpful to add that the fairness metrics and datasets you use are just intended to showcase your theoretical findings – but that you’re not implying that these fairness metrics are appropriate in the contexts that you use them in.
> >
> > ---
> > Thanks again for your thoughtful reply!

---

> > > ### Author Response · Authors · 2025-08-04
> > >
> > > We sincerely thank you for your thoughtful engagement with our rebuttal and for taking the extra time to provide a follow-up response. We appreciate the opportunity to continue the discussion and will address each of your points below.
> > >
> > >
> > > > **Individual fairness feasibility in facial recognition**
> > >
> > > We do agree that defining a facial similarity metric in facial recognition task may end up in problematic domains like physiognomy and may suffer from ethical challenges. It is precisely because of these difficulties that our work, and much of the current literature in this area, focuses on *group fairness*, which is more clearly defined and feasible.
> > >
> > > > **Why demographic parity for gender classification?**
> > >
> > > Thank you for the clarification on the question and sorry if we misunderstood your initial question. As we understand, what you are referring to is a fundamental challenge in fairness: the tension between accuracy and demographic parity, which arises when the base rates of a class, like females, differ across sensitive attributes. The base rate is the proportion of a class within a subgroup (e.g., the percentage of individuals labeled 'female' within the Black sub-group). In our context (using FairFace dataset for gender classification while treating race as the sensitive attribute) if the base rates are different, i.e., the number of Black females is far less than the number of White females, and we impose demographic parity, this could result in losing accuracy. That being said, the reason we use demographic parity is threefold:
> > >
> > > **1.** We use FairFace dataset in gender classification task. Our analysis reveals that there is not a huge difference in the base rate for sensitive attributes, specifically for race. As an example, there are 757 black females and 773 East Asian females in the dataset. Or, as another example, there are 799 black males and 777 East Asian males. So, there is not a huge difference in base rates.
> > >
> > > **2.** The reason demographic parity remains a widely used metric in different applications, including gender classification, despite this potential conflict with accuracy, is that its application often reflects policies and ethical goals. In many real-world scenarios, policy makers and stakeholders may decide that the outcomes should be independent of a sensitive attribute, even if that requires deviating from a model that perfectly replicates the imbalances present in historical data.
> > >
> > > **3.** Using demographic parity in gender classification task is common in the recent literature. As an example, in [1] and [2], the authors use demographic parity in gender classification tasks.
> > >
> > > [1] Chai, Junyi, Taeuk Jang, and Xiaoqian Wang. "Fairness without demographics through knowledge distillation." NeurIPS (2022).
> > >
> > > [2] Chai, Junyi, and Xiaoqian Wang. "Self-supervised fair representation learning without demographics." NeurIPS (2022).
> > >
> > > That being said, our paper's primary goal is not to prescribe demographic parity as the definitive or "correct" fairness metric for this or any task. Rather, our work is an investigation into a core ML component: the optimizer. To conduct this investigation rigorously, we must evaluate its impact on the established and standard metrics that the community uses. Hope that this time we correctly understood your concern.
> > >
> > > > **Change in optimizer to be used with other fairness-enhancing methods**
> > >
> > > Thank you for reiterating this important point. You are correct that the choice of any fairness metric is highly context-dependent, and your example regarding the COMPAS dataset perfectly makes sense. The purpose of our empirical evaluation is not to prescribe a specific fairness solution for these real-world tasks. Rather, our goal is to use these standard and widely recognized benchmarks to provide a clear and comparable validation of our theoretical findings.
> > >
> > > To ensure this is perfectly clear to the reader, we will add the following statement to the experimental setup section of our paper:
> > >
> > > "We note that the application of any particular group fairness metric is a context-dependent decision that requires careful ethical consideration. Our use of these standard datasets and fairness criteria is intended to empirically validate our theoretical findings on optimizer behavior across established benchmarks. This should not be interpreted as an endorsement of using these specific metrics for these particular real-world applications."
> > >
> > > Thank you again for this crucial feedback.

---

> > > > ### Comment · Reviewer_htpx · 2025-08-05
> > > >
> > > > Thank you for your thoughtful answer! This clarifies all my open questions (and to be honest I think there was a fault in my logic on point 2, so thanks for clarifying that).

---

> > > > > ### Author Response · Authors · 2025-08-05
> > > > >
> > > > > Thank you for the kind follow-up! We’re very happy to hear that our response was helpful. We are sincerely grateful for your thoughtful comments and strong support for our work throughout this entire process. Thank you again!

---

### Note · Authors · 2025-08-12

We thank the Reviewers and the Area Chair for their time and for a constructive review process.

We are encouraged that prior to the rebuttal, the reviewers highlighted several strengths of our work, including:

* Significance of the contribution

* Providing the provable fairness bounds

* Well-motivated introduction

* Extensive empirical validation across multiple datasets, tasks, and architectures.

During the discussion period, we worked to address all reviewer concerns. We are pleased that our efforts are well-received, resulting in one reviewer stating our rebuttal led to a **noteworthy contributions** and consequently raising their score. **All reviewers kindly acknowledged** that our responses clarified their questions.

To summarize, the following additions have been provided during the rebuttal, which will be incorporated into the final version:

* More generalized theorems: We relaxed the assumptions in Theorems 2 and 3 to the more challenging anisotropic noise setting.

* Confirmed generalizability beyond vision: We ran new experiments on a different modality (tabular data, using COMPAS and AdultCensus datasets) to demonstrate that our findings are not domain-specific.

* Extended analysis to optimizer variants: We conducted new experiments with AdamW, AdaBound, and SGD w/momentum optimizers to validate our theory's predictions about the central role of the second-moment normalization.

* Demonstrated real-world significance: We performed an additional experiment showing that the fairness advantage of adaptive optimizers is complementary and persists even when used along with the existing fairness-enhancing algorithms.

* Validated robustness to hyperparameters: We conducted a more comprehensive hyperparameter search to confirm our results are robust.

* Added context on fairness metrics: As suggested by the reviewer, we will add a note clarifying that our use of standard fairness metrics is only for the empirical validation of our claims about optimizers.

Thank you once again for reviewing our work and for providing such valuable feedback, which resulted in improving our paper.

---

### Decision · Program_Chairs · 2025-09-17

**Decision:**

Accept (spotlight)

**Comment:**

This paper investigates the under-explored relationship between optimization algorithms and group fairness in deep neural networks, demonstrating through stochastic differential equation analysis that adaptive optimizers like RMSProp and Adam consistently achieve fairer outcomes than SGD while maintaining comparable accuracy. The reviewers recognized the work's significant contributions, particularly its novel theoretical framework connecting optimizer dynamics to fairness outcomes and the practical insight that simply changing the optimizer can improve fairness without modifying model architecture or loss functions. During the rebuttal period, the authors effectively addressed all reviewer concerns by providing generalized theorems with relaxed assumptions and conducting experiments on tabular data to confirm generalizability beyond vision. While some reviewers initially questioned practical significance given existing fairness-aware algorithms, the authors convincingly demonstrated that optimizer choice serves as a fundamental, orthogonal mechanism that amplifies the effectiveness of existing fairness interventions rather than replacing them. Given the paper's novel theoretical insights into an overlooked aspect of fairness, strong empirical validation, and thorough responses addressing all reviewer concerns, I recommend accepting this paper for NeurIPS 2025.